# Feature selection and low test error in shallow low-rotation ReLU networks

**Matus Telgarsky**
`<mjt@illinois.edu>`

## Abstract

This work establishes low test error of gradient flow (GF) and stochastic gradient descent (SGD) on two-layer ReLU networks with standard initialization scale, in three regimes where key sets of weights rotate little (either naturally due to GF and SGD, or due to an artificial constraint), and making use of margins as the core analysis technique. The first regime is near initialization, specifically until the weights have moved by $\mathcal{O}(\sqrt{m})$, where $m$ denotes the network width, which is in sharp contrast to the $\mathcal{O}(1)$ weight motion allowed by the Neural Tangent Kernel (NTK); here it is shown that GF and SGD only need a network width and number of samples inversely proportional to the NTK margin, and moreover that GF attains at least the NTK margin itself and in particular escapes bad KKT points of the margin objective, whereas prior work could only establish nondecreasing but arbitrarily small margins. The second regime is the Neural Collapse (NC) setting, where data lies in well-separated groups, and the sample complexity scales with the number of groups; here the contribution over prior work is an analysis of the entire GF trajectory from initialization. Lastly, if the inner layer weights are constrained to change in norm only and can not rotate, then GF with large widths achieves globally maximal margins, and its sample complexity scales with their inverse; this is in contrast to prior work, which required infinite width and a tricky dual convergence assumption.

## 1 Introduction

A key promise of deep learning is *automatic feature learning*: standard gradient methods are able to adjust network parameters so that lower layers become meaningful feature extractors, which in turn implies low sample complexity. As a running illustrative (albeit technical) example throughout this work, in the *2-sparse parity* problem (cf. Figure 1), networks near initialization require $d^2/\epsilon$ samples to achieve $\epsilon$ test error, whereas powerful optimization techniques are able to learn more compact networks which need only $d/\epsilon$ samples (Wei et al., 2018). It is not clear how to establish this improved feature learning ability with a standard gradient-based optimization method; for example, despite the incredible success of the *Neural Tangent Kernel (NTK)* in proving various training and test error guarantees (Jacot et al., 2018; Du et al., 2018b; Allen-Zhu et al., 2018; Zou et al., 2018; Arora et al., 2019; Li & Liang, 2018; Ji & Telgarsky, 2020b; Oymak & Soltanolkotabi, 2019), ultimately the NTK corresponds to learning with frozen initial random features.

The goal of this work is to establish low test error from random initialization in an intermediate regime where parameters of individual nodes do not rotate much, however their change in norm leads to *selection* of certain pre-existing features. This perspective is sufficient to establish the best known sample complexities from random initialization in a variety of scenarios, for instance matching the $d^2/\epsilon$ within-kernel sample complexity with a computationally-efficient stochastic gradient descent (SGD) method, and the beyond-kernel $d/\epsilon$ sample complexity with an inefficient gradient flow (GF) method. The different results are tied together through their analyses, which establish not merely low training error but *large margins*, a classical approach to low sample complexity within overparameterized models (Bartlett, 1996). These results will use standard gradient methods from standard initialization, which is in contrast to existing works in feature learning, which adjusts the optimization method in some way (Shi et al., 2022; Wei et al., 2018), most commonly by training the inner layer for only one iteration (Daniely & Malach, 2020; Abbe et al., 2022; Barak et al., 2022;

Damian et al., 2022), and typically not beating the within-kernel $d^2/\epsilon$ sample complexity on the 2-sparse parity problem (cf. Table 1).

**Contributions.** There are four high-level contributions of this work. The first two consider networks of reasonable width (e.g., $\mathcal{O}(d^2)$ for 2-sparse parity), and are the more tractable of the four. In these results, the network parameters can move up to $\mathcal{O}(\sqrt{m})$, where $m$ is the width of the network; this is in sharp contrast to the NTK, where weights can only move by $\mathcal{O}(1)$. The performance of these first two results is measured in terms of the *NTK margin* $\gamma_{\text{ntk}}$, a quantity formally defined in Assumption 1.2. These first two contributions are as follows.

1. **Non-trivial margin KKT points.** Prior work established that features converge in a strong sense: features and parameters converge to a *KKT point* of a natural margin objection (cf. Section 1.1, (Lyu & Li, 2019; Ji & Telgarsky, 2020a)). Those works, however, left open the possibility that the limiting KKT point is arbitrarily bad; instead, Theorem 2.1 guarantees that the limiting GF margin is at least $\gamma_{\text{ntk}}/4096$, where $\gamma_{\text{ntk}}$ is a *distribution-dependent constant*.

2. **Simultaneous low test error and low computational complexity.** Replacing GF with SGD in the preceding approach leads to a computationally efficient method. Applying the resulting guarantees in Theorem 2.3 to the 2-sparse parity problem yields, as detailed in Table 1, a method which saves a factor $d^8$ against prior work with sample complexity $d^2/\epsilon$, and a factor $1/\epsilon$ in computation against work with sample complexity $d^4/\epsilon^2$. Moreover, Theorem 2.3 guarantees that the first gradient step moves parameters by $\sqrt{m}$ and formally exits the NTK.

The second two high-level contributions require intractable widths (e.g., $2^d$), but are able to achieve much better *global* margins $\gamma_{\text{gl}}$, which as detailed in Sections 1.1 and 1.2, were previously only possible under strong assumptions or unrealistic algorithmic modifications.

3. **Neural collapse.** Theorem 3.2 establishes low sample complexity in the *neural collapse (NC)* regime (Papyan et al., 2020), where data are organized in well-separated clusters of common label. By contrast, prior work did not analyze gradient methods from initialization, but instead the relationship between various optimality conditions (Papyan et al., 2020; Yaras et al., 2022; Thrampoulidis et al., 2022). The method of proof is to establish global margin maximization of GF; by contrast, for any type of data, this was only proved in the literature with strong assumptions and modified algorithms (Wei et al., 2018; Chizat & Bach, 2020; Lyu et al., 2021).

4. **Global margin maximization for rotation-free networks.** To investigate what could be possible, Theorem 3.3 establishes global margin maximization with GF under a restriction that the inner weights can only change in norm, and can not rotate; this analysis suffices to achieve $d/\epsilon$ sample complexity on 2-sparse parity, as in Table 1, and the low-rotation assumption is backed by preliminary empirical evidence in Figure 2.

As purely technical contributions, this work provides new tools to analyze low-width networks near initialization (cf. Lemmas B.4 and C.4), a new versatile generalization bound technique (cf. Lemma C.5), and a new potential function technique for global margin maximization far from initialization (cf. Lemma B.7 and applications thereof).

This introduction concludes with notation and related work, Section 2 collects the KKT point and low computation guarantees, Section 3 collects the global margin guarantees, Section 4 provides concluding remarks and open problems, and the appendices contain full proofs and additional technical discussion.

## 1.1 NOTATION

**Architecture and initialization.** With the exception of Theorem 3.3, the architecture will be a 2-layer ReLU network of the form $x \mapsto F(x; W) = \sum_j a_j \sigma(v_j^\mathsf{T} x) = a^\mathsf{T} \sigma(Vx)$, where $\sigma(z) = \max\{0, z\}$ is the ReLU, and where $a \in \mathbb{R}^m$ and $V \in \mathbb{R}^{m \times d}$ have initialization scale roughly matching `pytorch` defaults: $a \sim \mathcal{N}_m/m^{1/4}$ ($m$ iid Gaussians with variance $1/\sqrt{m}$) and $V \sim \mathcal{N}_{m \times d}/\sqrt{d\sqrt{m}}$ ($m \times d$ iid Gaussians with variance $1/(d\sqrt{m})$); in contrast with `pytorch`, the layers are approximately balanced. These parameters $(a, V)$ will be collected into a tuple $W =$

$(a, V) \in \mathbb{R}^m \times \mathbb{R}^{m \times d} \equiv \mathbb{R}^{m \times (d+1)}$, and for convenience per-node tuples $w_j = (a_j, v_j) \in \mathbb{R} \times \mathbb{R}^d \equiv \mathbb{R}^{d+1}$ will often be used as well.

Given a pair $(x, y)$ with $x \in \mathbb{R}^d$ and $y \in \{\pm 1\}$, the *prediction* or *unnormalized margin mapping* is $p(x, y; W) = yF(x; W) = ya^\intercal\sigma(Vx)$; when examples $((x_i, y_i))_{i=1}^n$ are available, a simplified notation $p_i(W) := p(x_i, y_i; W)$ is often used, and moreover define a single-node variant $p_i(w_j) := y_i a_j \sigma(v_j^\intercal x_i)$. Throughout this work, $\|x\| \leq 1$, and unmarked norms are Frobenius norms.

**SGD and GF.** The loss function $\ell$ will be either the exponential loss $\ell_{\exp}(z) := \exp(-z)$, or the logistic loss $\ell_{\log}(z) := \ln(1 + \exp(-z))$; the corresponding empirical risk $\widehat{\mathcal{R}}$ is

$$\widehat{\mathcal{R}}(p(W)) := \frac{1}{n} \sum_{i=1}^n \ell(p_i(W)),$$

which used $p(W) := (p_1(W), \ldots, p_n(W)) \in \mathbb{R}^n$. The two descent methods are

$$W_{i+1} := W_i - \eta \hat{\partial}_W \ell(p_i(W_i)), \qquad \text{stochastic gradient descent (SGD),} \qquad (1.1)$$

$$\dot{W}_t := \frac{\mathrm{d}}{\mathrm{d}t} W_t = -\bar{\partial}_W \widehat{\mathcal{R}}(p(W_t)), \qquad \text{gradient flow (GF),} \qquad (1.2)$$

where $\hat{\partial}$ and $\bar{\partial}$ are appropriate generalizations of subgradients for the present nonsmooth nonconvex setting, detailed as follows. For SGD, $\hat{\partial}$ will denote any valid element of the *Clarke differential* (i.e., a measurable selection); for example, $\hat{\partial}F(x; W) = \left(\sigma(Vx), \sum_j a_j \sigma'(v_j^\intercal x)e_j x^\intercal\right)$, where $e_j$ denotes the $j$th standard basis vector, and $\sigma'(v_j^\intercal x_i) \in [0, 1]$ is chosen in some consistent and measurable way, for instance as chosen by `pytorch`. For GF, $\bar{\partial}$ will denote the unique minimum norm element of the Clarke differential; typically, GF is defined as a differential inclusion, which agrees with this minimum norm Clarke flow almost everywhere, but here the minimum norm element is equivalently used to define the flow. Details of Clarke differentials and corresponding chain rules are differed to the now-extensive literature for their use in margin analyses (Lyu & Li, 2019; Ji & Telgarsky, 2020a; Lyu et al., 2021).

Due to clutter, the time indices $t$ (written as $W_t$ or $W(t)$) will often be dropped.

**Margins.** To develop the margin notion, first note that that $F$ and $p_i$ are *2-homogeneous* in $W$, meaning $F(x; cW) = ca^\intercal\sigma(cVx) = c^2 F(x; W)$ for any $x \in \mathbb{R}^d$ and $c \geq 0$ (and $p_i(cW) = c^2 p_i(W)$). It follows that $F(x; W) = \|W\|^2 F(x; W/\|W\|)$, and thus $F$ and $p_i$ scale quadratically in $\|W\|$, and it makes sense to define a *normalized* prediction mapping $\widetilde{p}_i$ and margin $\gamma$ as

$$\widetilde{p}_i(W) := \frac{p_i(W)}{\|W\|^2}, \qquad \gamma(W) := \frac{\min_i p_i(W)}{\|W\|^2} = \min_i \widetilde{p}_i(W).$$

Due to nonsmoothness, $\gamma$ can be hard to work with, thus, following Lyu & Li (2019), define the *smoothed margin* $\widetilde{\gamma}$ and the *normalized smoothed margin* $\mathring{\gamma}$ as

$$\widetilde{\gamma}(W) := \ell^{-1}\left(n\widehat{\mathcal{R}}(W)\right) = \ell^{-1}\left(\sum_i \ell(p_i(W))\right), \qquad \mathring{\gamma}(W) := \frac{\widetilde{\gamma}(W)}{\|W\|^2},$$

where a key result is that $\mathring{\gamma}$ is eventually nondecreasing (Lyu & Li, 2019). These quantities may look complicated and abstract, but note for $\ell_{\exp}$ that $\widetilde{\gamma}(W) := -\ln\left(\sum_i \exp(-p_i(W))\right)$. An interesting technical consideration is that normalization by $\|W\|^2$ can be replaced by $\|a\| \cdot \|V\|$ or $\sum_j \|a_j v_j\|$, as appears throughout the proofs of Theorems 2.1, 3.2 and 3.3.

Corresponding to these definitions, the *global* max margin assumption is that *some* shallow network can achieve a good margin almost surely over the distribution.

**Assumption 1.1.** There exists $\gamma_{\text{gl}} > 0$ and parameters $((\alpha_k, \beta_k))_{k=1}^r$ with $\|\alpha\|_1 \leq 1$ and $\|\beta_k\|_2 = 1$ so that almost surely over the draw of any pair $(x, y)$, then $y \sum_k \alpha_k \sigma(\beta_k^\intercal x) \geq \gamma_{\text{gl}}$. $\diamondsuit$

The $\ell_1$ norm on $\alpha$ is due to 2-homogeneity: for any 2-homogeneous generalized activation $\phi$ and parameters $(w_j)_{j=1}^m$ organized into matrix $W \in \mathbb{R}^{m \times (d+1)}$, then $\sum_j \phi(w_j^\intercal x)/\|W\|^2 =$

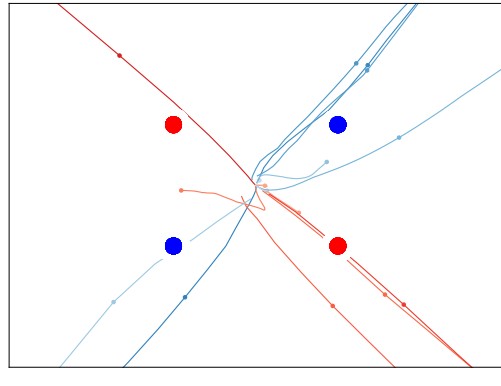
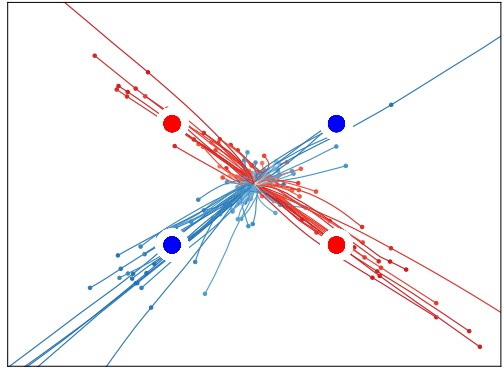

(a) Trajectories $(|a_j|v_j)_{j=1}^m$ with $m = 16$.     (b) Trajectories $(|a_j|v_j)_{j=1}^m$ with $m = 256$.

Figure 1: Two-dimensional projection of $n = 64$ samples drawn from the 2-sparse parity distribution in $d = 20$ dimensions (cf. Proposition 1.3), with red and blue circles respectively denoting negative and positive examples. Red paths correspond to trajectories $|a_j|v_j$ across time with $a_j < 0$, whereas blue paths have $a_j > 0$.

$\sum_j (\|w_j\|/\|W\|)^2 \phi((w_j/\|w_j\|)^\intercal x)$, and in particular $\sum_j \|w_j\|^2/\|W\|^2 = 1$. This use of the $\ell_1$ norm is standard in works studying global margin maximization, see for instance (Chizat & Bach, 2020, Proposition 12, optimality conditions). Near initialization, the features have not changed much, and it is therefore reasonable to consider a second margin definition as linear predictors on top of the random initial features.

**Assumption 1.2.** There exists $\gamma_{\text{ntk}} > 0$ and a weight mapping $\theta : \mathbb{R}^{d+1} \to \mathbb{R}^{d+1}$ with $\theta(w) = 0$ whenever $\|w\| \geq 2$ and $\|\theta(w)\| \leq 2$ otherwise, so that almost surely over the draw of $(x, y)$, then $\mathbb{E}_{w \sim \mathcal{N}_\theta} \langle \theta(w), \bar{\partial}_w p(x; w) \rangle \geq \gamma_{\text{ntk}}$, where $w = (a, v) \sim \mathcal{N}_\theta$ means $a \sim \mathcal{N}$ and $v \sim \mathcal{N}_d/\sqrt{d}$, and $p(x, y; w) = p(x, y; (a, v)) = ya\sigma(v^\intercal x)$ as before. $\diamondsuit$

Assumption 1.2 may seem overly technical, but by taking an expectation over initial weights, it can not only be seen as an infinite-width linear predictor over the initial features, but moreover it is a deterministic condition and not a random variable depending on the sampled weights. This assumption was originally presented by Nitanda & Suzuki (2019) and later used in (Ji & Telgarsky, 2020b); a follow-up work with similar proof techniques made the choice of using a finite-width assumption which is a random variable (Chen et al., 2019).

The 2-sparse parity problem, finally formally defined in Proposition 1.3 as follows and depicted in Figure 1, allows significantly different estimates for Assumption 1.1 and Assumption 1.2. A notable feature of this version, in contrast to prior work, is that only the support of the distribution matters.

**Proposition 1.3.** *Consider any* 2-*sparse parity data distribution: inputs are supported on* $H_d := \{\pm 1/\sqrt{d}\}^d$, *and for any* $x \in H_d$, *the label is the product of two fixed coordinates* $y := d x_a x_b$ *with* $a \neq b$. *Then Assumption 1.2 holds with* $\gamma_{\text{ntk}} \geq \frac{1}{50d}$, *and Assumption 1.1 holds with* $\gamma_{\text{gl}} \geq \frac{1}{\sqrt{8d}}$.

Lastly, following (Lyu & Li, 2019) but simplified for the 2-homogeneous case, given examples $((x_i, y_i))_{i=1}^n$, then parameters $W = (a, V)$ are a *KKT point* if there exist *Lagrange multipliers* $(\lambda_1, \ldots, \lambda_n)$ with $\lambda_i \geq 0$ and $\lambda_i > 0$ only if $p_i(W) = 1$, and moreover, for every $j$, then $a_j = \sum_i \lambda_i y_i \sigma(v_j^\intercal x_i)$ and $v_j \in a_j \sum_i \lambda_i y_i \partial_v \sigma(v_j^\intercal x_i)$, where $\partial_v$ denotes the Clarke differential with respect to $v_j$. Call $W$ a *KKT direction* if there exists a scalar $r > 0$ so that $rW$ is a KKT point. Lastly, the margin of a KKT point $W$ is $1/\|W\|^2$, and the margin of a KKT direction is the margin of the corresponding KKT point; further details on these notions are deferred to Appendix B.1.

### 1.2   FURTHER RELATED WORK

**Margin maximization.** The concept and analytical use of margins in machine learning originated in the classical perceptron convergence analysis of Novikoff (1962). The SGD analysis in Theorem 2.3, as well as the training error analysis in Lemma C.3 were both established with a variant

| Reference | Algorithm | Technique | $m$ | $n$ | $t$ |
|---|---|---|---|---|---|
| (Ji & Telgarsky, 2020b) | SGD | perceptron | $d^8$ | $d^2/\epsilon$ | $d^2/\epsilon$ |
| Theorem 2.3 | SGD | perceptron | $d^2$ | $d^2/\epsilon$ | $d^2/\epsilon$ |
| (Barak et al., 2022) | 2-phase SGD | correlation | $\mathcal{O}(1)$ | $d^4/\epsilon^2$ | $d^2/\epsilon^2$ |
| (Wei et al., 2018) | WF+noise | margin | $\infty$ | $d/\epsilon$ | $\infty$ |
| (Chizat & Bach, 2020) | WF | margin | $\infty$ | $d/\epsilon$ | $\infty$ |
| Theorem 3.3 | scalar GF | margin | $d^{d/2}$ | $d/\epsilon$ | $\infty$ |

Table 1: Performance on 2-sparse parity by a variety of works, loosely organized by technique; see Section 1.2 for details. Briefly, $m$ denotes width, $n$ denotes *total* number of samples (across all iterations), and $t$ denotes the number of algorithm iterations. Overall, the table exhibits many tradeoffs, and there is no single best method.

of the perceptron proof; similar perceptron-based proofs appeared before (Ji & Telgarsky, 2020b; Chen et al., 2019), however they required width $1/\gamma_{\mathrm{ntk}}^8$, unlike the $1/\gamma_{\mathrm{ntk}}^2$ here, and moreover the proofs themselves were in the NTK regime, whereas the proof here is not.

Works focusing on the *implicit margin maximization* or *implicit bias* of descent methods are more recent. Early works on the coordinate descent side are (Schapire et al., 1997; Zhang & Yu, 2005; Telgarsky, 2013); the proof here of Lemma C.4 uses roughly the proof scheme in (Telgarsky, 2013). More recently, margin maximization properties of gradient descent were established, first showing global margin maximization in linear models (Soudry et al., 2017; Ji & Telgarsky, 2018b), then showing nondecreasing *smoothed* margins of general homogeneous networks (including multi-layer ReLU networks) (Lyu & Li, 2019), and the aforementioned *global* margin maximization result for 2-layer networks under dual convergence and infinite width (Chizat & Bach, 2020). The potential functions used here in Theorems 3.2 and 3.3 use ideas from (Soudry et al., 2017; Lyu & Li, 2019; Chizat & Bach, 2020), but also the shallow linear and deep linear proofs of Ji & Telgarsky (2019; 2018a).

**Feature learning.** There are many works in feature learning, a few also carrying explicit guarantees on 2-sparse parity are summarized in Table 1. An early work with high relevance to the present work is (Wei et al., 2018), which in addition to establishing that the NTK requires $\Omega(d^2/\epsilon)$ samples whereas $\mathcal{O}(d/\epsilon)$ suffice for the global maximum margin solution, also provided a noisy Wasserstein Flow (WF) analysis which achieved the maximum margin solution, albeit using noise, infinite width, and continuous time to aid in local search. The global maximum margin work of Chizat & Bach (2020) was mentioned before, and will be discussed in Section 3. The work of Barak et al. (2022) uses a two phase algorithm: the first step has a large minibatch and effectively learns the support of the parity in an unsupervised manner, and thereafter only the second layer is trained, a convex problem which is able to identify the signs within the parity; as in Table 1, this work stands alone in terms of the narrow width it can handle. The work of (Abbe et al., 2022) uses a similar two-phase approach, and while it can not learn precisely the parity, it can learn an interesting class of "staircase" functions, and presents many valuable proof techniques. Another work which operates in two phases and can learn an interesting class of functions is the recent work of (Damian et al., 2022); while it can not handle 2-sparse parity explicitly, it can handle the Hermite polynomial analog (product of two hermite polynomials). Other interesting feature learning works are (Shi et al., 2022; Bai & Lee, 2019; Allen-Zhu & Li, 2020).

## 2  LOW TEST ERROR WITH MODEST-WIDTH NETWORKS

This section states the aforementioned results for networks of width $\Omega(1/\gamma_{\mathrm{ntk}}^2)$, which can be small: as provided by Proposition 1.3, this width is $\Omega(d^2)$ for the 2-sparse parity problem. This section will first give guarantees for GF, establishing via Theorem 2.1 and Corollary 2.2 that non-trivial KKT points are achieved. Similar ideas will then be used to give a fully-tractable SGD approach in Theorem 2.3. To start, here is the low test error and large margin guarantee for GF.

**Theorem 2.1.** *Suppose the data distribution satisfies Assumption 1.2 for some $\gamma_{\mathrm{ntk}} > 0$, and the GF curve $(W_s)_{s \geq 0}$ uses $\ell \in \{\ell_{\exp}, \ell_{\log}\}$ on an architecture of width $m \geq \left(\frac{640 \ln(n/\delta)}{\gamma_{\mathrm{ntk}}}\right)^2$. Then, with*

*probability at least* $1 - 15\delta$, *there exists* $t$ *with* $\|W_t - W_0\| = \gamma_{\mathrm{ntk}}\sqrt{m}/32$ *so that, for all* $s \geq t$,

$$\frac{\widetilde{\gamma}(W_s)}{\|a_s\| \cdot \|V_s\|} \geq \frac{\gamma_{\mathrm{ntk}}}{2048} \qquad \text{and} \qquad \Pr[p(x, y; W_s) \leq 0] \leq \mathcal{O}\left(\frac{\ln(n)^3}{n\gamma_{\mathrm{ntk}}^2} + \frac{\ln\frac{1}{\delta}}{n}\right),$$

*and moreover* $\liminf_{s \to \infty} \gamma(W_s) = \liminf_{s \to \infty} \frac{\min_i p_i(W_s)}{2\|a_s\| \cdot \|V_s\|} \geq \frac{\gamma_{\mathrm{ntk}}}{4096}$.

Before sketching the proof, one interesting comparison is to a leaky ReLU convergence analysis on a restricted form of linearly separable data due to Lyu et al. (2021). That work, through an extremely technical and impressive analysis, establishes convergence to a solution which is equivalent to the best linear predictor. By contrast, while the work here does not recover that analysis, since $\gamma_{\mathrm{ntk}} = \Omega(\gamma_0)$ where $\gamma_0$ is the linear separability margin (cf. Proposition B.1), then the margin and sample complexity achieved here are within a constant factor of those in (Lyu et al., 2021), but via a simpler and more general analysis (dropping the additional data and initialization conditions).

The proof of Theorem 2.1 is provided in full in the appendices, but has the following key components. The main tool powering all results in this section, Lemma B.4, can be roughly stated as follows: gradients at initialization are aligned with a fixed good parameter direction $\bar{\theta} \in \mathbb{R}^{m \times (d+1)}$ with $\|\bar{\theta}\| \leq 2$, meaning $\langle \bar{\theta}, \bar{\partial}p_i(W_0) \rangle \geq \gamma_{\mathrm{ntk}}\sqrt{m}/2$, and moreover nearly the same inequality holds with $W_0$ replaced by any $W \in \mathbb{R}^{m \times (d+1)}$ with $\|W - W_0\| \leq \gamma_{\mathrm{ntk}}\sqrt{m}$. This is a form of Polyak-Łojasiewicz inequality, and guides the gradient flow in a good direction, and is used in a strengthened form to obtain an empirical risk guarantee for GF (large margins and low test error will be discussed shortly). While a version of this inequality has appeared in prior work (Ji & Telgarsky, 2020b), despite adaptations to multi-layer cases (Chen et al., 2019), all prior work had a width dependence of $1/\gamma_{\mathrm{ntk}}^8$; many careful refinements here lead to the smaller width $1/\gamma_{\mathrm{ntk}}^2$. Overall, as in (Ji & Telgarsky, 2020b), the proof technique is based on the classical perceptron analysis, and the width requirement here matches the width needed by perceptron with frozen initial features.

The proof then continues by establishing large margins, and then by applying a large-margin generalization bound. The margin analysis, surprisingly, is a 2-homogeneous adaptation of a large-margin proof technique for coordinate descent (Telgarsky, 2013), and uses the preceding empirical risk guarantee for a warm start. The generalization analysis follows a new proof technique and may be of independent interest, and appears in full in Lemma C.6. An interesting detail in these proofs is that the margins behave better when normalized with the nonstandard choice $\|a\| \cdot \|V\|$.

As discussed above, Theorem 2.1 is complemented by Corollary 2.2, which establishes that GF can sometimes escape bad KKT points.

**Corollary 2.2.** *Let* $\gamma_0 \in (0, 1/4)$ *be given, and consider the uniform distribution on the two points* $z_1 = (\gamma_0, +\sqrt{1 - \gamma_0^2})$ *and* $z_2 = (\gamma_0, -\sqrt{1 - \gamma_0^2})$ *with common label* $y = +1$. *With probability at least* $1 - 2^{1-n}$ *over an iid draw from this distribution, for any width* $m$, *the choice* $a_j = 1$ *and* $v_j = (1, 0)$ *for all* $j$ *is a KKT direction with margin* $\gamma_0/2$. *On the other hand, with probability at least* $1 - 15\delta$, *GF with loss* $\ell_{\exp}$ *or* $\ell_{\log}$ *on an iid sample of size* $n$ *from this data distribution using width at least* $m \geq 2^{50}\ln(n/\delta)^2$ *converges to a KKT direction with margin at least* $2^{-27}$.

Summarizing, GF achieves at least constant margin $2^{-27}$, whereas the provided KKT point achieves the arbitrarily small margin $\gamma_0/2$; as such, choosing any $\gamma_0 < 2^{-26}$ guarantees that GF converges to a nontrivial KKT point. While this construction may seem artificial, it is a simplified instance of the *neural collapse* constructions in Section 3, and by contrast with the results there, is achievable with reasonably small widths.

To close this section, the corresponding SGD guarantee is as follows. This result gives a fully tractable method, and appears in Table 1. Notably, this proof can not handle the exponential loss, since gradient norms do not seem to concentrate.

**Theorem 2.3.** *Suppose the data distribution satisfies Assumption 1.2 for some* $\gamma_{\mathrm{ntk}} > 0$, *let time* $t$ *be given, and suppose width* $m$ *and step size* $\eta$ *satisfy*

$$m \geq \left(\frac{64\ln(t/\delta)}{\gamma_{\mathrm{ntk}}}\right)^2 \qquad \text{and} \qquad \eta \in \left[\frac{\gamma_{\mathrm{ntk}}}{10\sqrt{m}}, \frac{\gamma_{\mathrm{ntk}}^2}{6400}\right].$$

*Then, with probability at least $1 - 8\delta$, the SGD iterates $(W_s)_{s \leq t}$ with logistic loss $\ell = \ell_{\log}$ satisfy*

$$\min_{s < t} \Pr\left[p(x, y; W_s) \leq 0\right] \leq \frac{8 \ln(1/\delta)}{t} + \frac{2560}{t\gamma_{\text{ntk}}^2}, \qquad \text{(test error bound)},$$

$$\max_{s < t} \|W_s - W_0\| \leq \frac{80\eta\sqrt{m}}{\gamma_{\text{ntk}}}, \qquad \text{(norm upper bound)},$$

$$\|W_1 - W_0\| \geq \frac{\eta\gamma_{\text{ntk}}\delta^4\sqrt{m}}{8}, \qquad \text{(norm lower bound)}.$$

A notable characteristic is exiting the NTK: choosing the largest allowed step size $\eta := \gamma_{\text{ntk}}^2/6400$, it follows that $\|W_1 - W_0\| \geq \delta^4\gamma_{\text{ntk}}^3\sqrt{m}/51200$, whereas the NTK regime only permits $\|W_t - W_0\| = \mathcal{O}(1)$. Of course, despite exiting the NTK, this sample complexity is still measured in terms of $\gamma_{\text{ntk}}$, suggesting many opportunities for future work.

A few remarks on the proof are as follows. Interestingly, it is much shorter than the GF proof, as it mainly needs to replicate GF's empirical risk guarantee, and then apply a short martingale concentration argument. A key issue, however, is the large squared gradient norm term, which is the source of the large lower bound on $\|W_1 - W_0\|$. A typical optimization analysis technique is to swallow this term by scaling the step size with $1/\sqrt{t}$ or $1/\sqrt{m}$, but here a constant step size is allowed. Instead, controlling the term is possible again using nuances of the perceptron proof technique, which controls the term $\sum_{i<t} |\ell'(p_i(W_i))|$, which appears when these squared gradients are accumulated.

## 3 LOWER TEST ERROR WITH LARGE-WIDTH NETWORKS

This section provides bounds which are more ambitious in terms of test error, but pay a big price: the network widths will be exponentially large, and either the data or the network architecture will have further conditions. Still, these settings will both be able to achieve globally maximal margins, and for instance lead to the improved $d/\epsilon$ sample complexity in Table 1.

### 3.1 NEURAL COLLAPSE (NC)

The *Neural Collapse (NC)* setting partitions data into well-separated groups (Papyan et al., 2020); these groups form narrow cones which meet at obtuse angles.

**Assumption 3.1.** There exist $((\alpha_k, \beta_k))_{k=1}^r$ with $\|\beta_k\| = 1$ and $\alpha_k \in \{\pm 1/r\}$ and $\gamma_{\text{nc}} > 0$ and $\epsilon \in (0, \gamma_{\text{nc}})$ so that almost surely for any $(x, y)$, for each $(\alpha_k, \beta_k)$ exactly one of the following hold:

- either $x$ lies in a cone around $\beta_k$, meaning $\text{sgn}(\alpha_k)\beta_k^\mathsf{T} xy \geq \gamma_{\text{nc}}$ and $\frac{\|(I - \beta_k\beta_k^\mathsf{T})x\|}{|\beta_k^\mathsf{T} x|} \leq \sqrt{\epsilon/2}$;

- or $x$ is bounded away from the cone around $\beta_k$, meaning $\text{sgn}(\alpha_k)\beta_k^\mathsf{T} xy \leq -\epsilon$. $\diamondsuit$

It follows that Assumption 3.1 implies Assumption 1.1 with margin $\gamma_{\text{gl}} \geq \gamma_{\text{nc}}/r$, but the condition is quite a bit stronger. The corresponding GF result is as follows.

**Theorem 3.2.** *Suppose the data distribution satisfies Assumption 3.1 for some $(r, \gamma_{\text{nc}}, \epsilon)$, and consider GF curve $(W_t)_{t \geq 0}$ with $\ell \in \{\ell_{\exp}, \ell_{\log}\}$ and width $m \geq 4\left(\frac{2}{\epsilon}\right)^{d-1}\ln\frac{r}{\delta}$. then, with probability at least $1 - \delta$, it holds for all large $t$ that*

$$\hat{\gamma}(W_t) \geq \frac{\gamma_{\text{nc}} - \epsilon}{2r} \qquad \text{and} \qquad \Pr\left[p(x, y; W_t) \leq 0\right] = \mathcal{O}\left(\frac{r^2 \ln(n)^3}{n(\gamma_{\text{nc}} - \epsilon)^2} + \frac{\ln\frac{1}{\delta}}{n}\right).$$

Before discussing the proof, here are a few remarks. Firstly, the standard NC literature primarily compares different optimality conditions and how they induce NC (Papyan et al., 2020; Yaras et al., 2022; Thrampoulidis et al., 2022); by contrast, Theorem 3.2 analyzes the behavior of a standard descent method on data following NC. Furthermore, Theorem 3.2 does not establish that GF necessarily converges to the NC solution, since Assumption 3.1 allows for scenarios where the globally maximal margin solution disagrees with NC. One such example is to take two points on the surface

of the sphere and which form an angle just beyond $\pi/2$; in this case, the globally maximal margin solution is equivalent to a single linear predictor, but Assumption 3.1 and Theorem 3.2 still apply. The similar construction in Corollary 2.2 took those two points and pushed their angle to be just below $\pi$, which made NC the globally maximal margin solution. Overall, the relationship between NC and the behavior of GF is quite delicate.

The proof of Theorem 3.2 hinges on a potential function $\Phi(W_t) := \frac{1}{4} \sum_k |\alpha_k| \ln \sum_j \phi_{k,j}(w_j) \|a_j v_j\|$, where $\phi_{k,j}(w_j)$ is near 1 when $v_j/\|v_j\| \approx \beta_k$, and 0 otherwise. This strange-looking potential has derivative scaling with $\gamma_{\mathrm{nc}}/r$ and a certain technical factor $\mathcal{Q}$; meanwhile, the derivative of $\ln \|W_t\|^2$ scales roughly with $\gamma(W_t)$ and that same factor $\mathcal{Q}$. Together, it follows by considering their difference that either $\gamma(W_t)$ must exceed $(\gamma_{\mathrm{nc}} - \epsilon)/r$, or mass must concentrate on the NC directions. This suffices to complete the proof, however there are many technical details; for instance, without the NC condition, data can pull gradients in bad directions and this particular potential function can become negative; in other words, the NC condition reduces rotation. The use of $\ln(\cdot)$ may seem bizarre, but it causes the gradient to be self-normalizing; similar self-normalizing ideas were used throughout earlier works on margins outside the NTK (Lyu & Li, 2019; Chizat & Bach, 2020; Ji & Telgarsky, 2020a). This discussion of $\Phi$ will resume after the proof of Theorem 3.3, which uses a similar construction.

One technical point of potentially independent interest is once again the use of $\|a_j v_j\|$ as a surrogate for $\|w_j\|^2$ (where $\|w_j\|^2 \geq 2\|a_j v_j\|$); this seems crucial in the proofs, and was also used in the proofs of Theorem 2.1, and also partially motivated the use of $|a_j| v_j$ when plotting the trajectories in Figure 1. While it is true that these quantities asymptotically balance (Du et al., 2018a), it takes quite a long time, and this more refined norm-like quantity is useful in early phases.

## 3.2 GLOBAL MARGIN MAXIMIZATION WITHOUT ROTATION

The final theorem will be on stylized networks where the inner layer is forced to not rotate. Specifically, the networks are of the form

$$x \mapsto \sum_j a_j \sigma(b_j v_j^{\mathsf{T}} x),$$

where $((a_j, b_j))_{j=1}^m$ are trained, but $v_j$ are fixed at initialization; the new scalar parameter $b_j$ is effectively the norm of $v_j$ (though it is allowed to be negative). As a further simplification, $a_j$ and $b_j$ are initialized to have the same norm; this initial balancing is common in many implicit bias proofs, but is impractical and constitutes a limitation to improve in future work. While these are clearly significant technical assumptions, we note firstly as in Figure 2 that low rotation seems to hold empirically, and moreover that the only other works establishing global margin maximization used either a significantly different algorithm with added gradient noise (Wei et al., 2018), or in the case of (Chizat & Bach, 2020), heavily relied upon infinite width (requiring weights to cover the sphere *for all times* $t$), and also a *dual convergence* assumption detailed in the appendices and circumvented here.

**Theorem 3.3.** *Suppose the data distribution satisfies Assumption 1.1 for some $\gamma_{\mathrm{gl}} > 0$ with reference architecture $((\alpha_k, \beta_k))_{k=1}^r$. Consider the architecture $x \mapsto \sum_j a_j \sigma(b_j v_j^{\mathsf{T}} x_i)$ where $((a_j(0), b_j(0)))_{j=1}^m$ are sampled uniformly from the two choices $\pm 1/m^{1/4}$, and $v_j(0)$ is sampled from the unit sphere (e.g., first $v_j' \sim \mathcal{N}_d$, then $v_j(0) := v_j'/\|v_j'\|$), and the width $m$ satisfies $m \geq 4 \left( \frac{4}{\gamma_{\mathrm{gl}}} \right)^{d-1} \ln \frac{r}{\delta}$. Then, with probability at least $1 - \delta$, for all large $t$, GF on $((a_j, b_j))_{j=1}^m$ with loss $\ell \in \{\ell_{\exp}, \ell_{\log}\}$ satisfies*

$$\mathring{\gamma}\left((a(t), b(t))\right) \geq \frac{\gamma_{\mathrm{gl}}}{2} \qquad \text{and} \qquad \Pr\left[p(x, y; (a(t), b(t))) \leq 0\right] = \mathcal{O}\left( \frac{\ln(n)^3}{n \gamma_{\mathrm{gl}}^2} + \frac{\ln \frac{1}{\delta}}{n} \right).$$

The proof strategy of Theorem 3.3 follows a simplification of the scheme from Theorem 3.2. First, due to the large width, for each $k$ there must exist a weight $j$ with $\mathrm{sgn}(b_j) v_j \approx \beta_k$. Second, since this inner layer can not rotate, we can reorder the weights so that simply $\mathrm{sgn}(b_k) v_k \approx \beta_k$, and define a simplified potential

$$\Phi(W_t) := \frac{1}{4} \sum_k |\alpha_k| \ln \left( a_k^2 + b_k^2 \right).$$

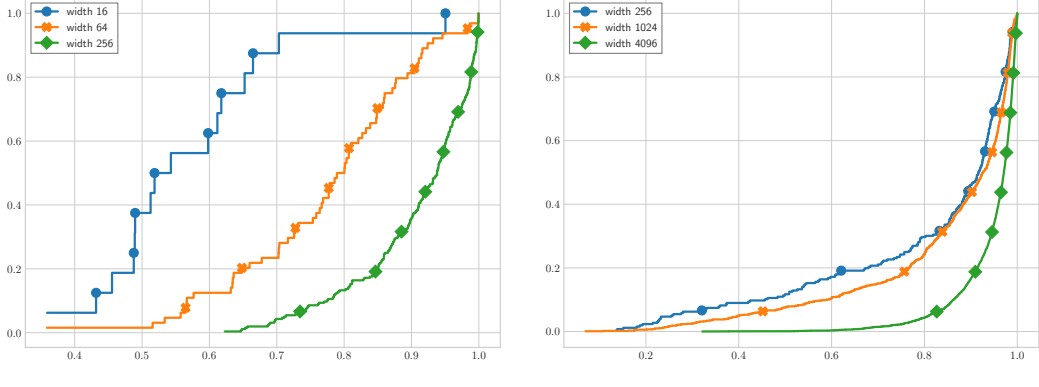

(a) Rotation for 2-sparse parity, $m \in \{16, 64, 256\}$.     (b) Rotation for `mnist`, $m \in \{256, 1024, 4096\}$.

Figure 2: Cumulative distribution functions (CDFs) for rotation on 2-sparse parity and `mnist` digits 3 vs 5, with three choices of width; both are run with small step size and full batch gradient descent until the empirical logistic risk is $1/n$, and the 2-sparse parity plots match the same invocation which gave Figure 1. To measure rotation, for any given width $m$, per-node rotations $\left\langle \frac{v_j(0)}{\|v_j(0)\|}, \frac{v_j(t)}{\|v_j(t)\|} \right\rangle$ are first calculated, and then treated as an empirical distribution, and their CDF is plotted. The overall trend is that as $m$ increases, rotation decreases. While this trend is consistent with the NTK, the rotations are still too large to allow an NTK-style analysis; further experimental details are in Appendix B.2.

As mentioned after the proof of Theorem 3.2, it's possible without Assumption 3.1 for data to pull weights in bad directions; that is ruled out here via the removal of rotation, and spiritually this situation is ruled out in the proof by Chizat & Bach (2020) via their dual convergence assumption.

## 4 CONCLUDING REMARKS AND OPEN PROBLEMS

Stated technically, this work provides a variety of settings where GF can achieve margins $\gamma_{\text{ntk}}$ and $\gamma_{\text{gl}}$ (and SGD, in one case, can achieve sample complexity and computation scaling nicely with $\gamma_{\text{ntk}}$), and whose behavior can be interpreted as GF and SGD *selecting* good features and achieving low test error. There are many directions for future work.

Figure 2 demonstrated low rotation with 2-sparse parity and `mnist`; can this be proved, thereby establishing Theorem 3.3 without forcing nodes to not rotate?

Looking to Table 1 for 2-sparse parity, the approaches here fail to achieve the lowest width; is there some way to achieve this with SGD and GF, perhaps even via margin analyses?

Theorem 2.3 and Theorem 2.1 achieve the same sample complexity for SGD and GF, but via drastically different proofs, the GF proof being weirdly complicated; is there a way to make the two more similar?

The approaches here are overly concerned with reaching a constant factor of the optimal margins; is there some way to achieve slightly worse margins with the benefit of reduced width and computation? More generally, what is the Pareto frontier of width, samples, and computation in Table 1?

The margin analysis here for the logistic loss, namely Theorem 2.1, requires a long warm start phase. Does this reflect practical regimes? Specifically, does good margin maximization and feature learning occur with the logistic loss in this early phase? This issue also appears in prior *linear* max margin works with the logistic loss.

The analyses here work best with $\sum_j \|a_j v_j\|$ in place of $\|W\|^2$; are there more natural choices, and can this choice be used in other aspects of deep learning analysis?

ACKNOWLEDGMENTS

MT thanks Fanny Yang, the Simons Institute, and the NSF (grant IIS-1750051).

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

## A  *Sincerest apologies to The Reader*

*Dear Reader,*

*I mismanaged time and my "plan" to prepare the camera ready in the final days was thwarted by a travel ambush. Please see the* `arXiv` *versions dating from after March 1, 2023 (hopefully available before March 31, 2023).*

*(This present version does not even address reviewer comments which were resolved in the rebuttal phase, updates will come shortly!)*

## B  Technical preliminaries

This first appendix section contains tools used throughout, extra background, and further experimental details.

### B.1  Further notation, background, and estimates of $\gamma_{ntk}$ and $\gamma_{gl}$

Firstly, in many of the proofs, it is useful to normalized parameters: define $\widetilde{v}_j := v_j/\|v_j\|$ and $\widetilde{a}_j := \operatorname{sgn}(a_j) := a_j/|a_j|$. Furthermore, write $\ell_i(W) = \ell(p_i(W))$ and $\ell_i'(W) := \ell'(p_i(W))$ are used; since $\ell_i'$ is negative, often $|\ell_i'|$ is written.

It is annoying to write $|\ell_i'|$ over and over, however, interestingly, these nonnegative derivatives can be transformed into a notion of *dual variable*, which will be used throughout the proofs. Concretely, define *dual variables* $(q_i)_{i=1}^n$

$$q := \nabla_p \ell^{-1}\left(\sum_i \ell(p_i)\right) = \frac{\nabla_p \sum_i \ell(p_i)}{\ell'(\ell^{-1}(\sum_i \ell(p_i)))} = \frac{\nabla_p \sum_i \ell(p)}{\ell'(\widetilde{\gamma}(p))},$$

which made use of the inverse function theorem. Correspondingly define $\mathcal{Q} := -\ell'(\widetilde{\gamma}(p))$, whereby $-\ell_i' = q_i\mathcal{Q}$; for the exponential loss, $\mathcal{Q} = \sum_i \exp(-p_i)$ and $\sum_i q_i = 1$, and while these quantities are more complicated for the logistic loss, they eventually satisfy $\sum_i q_i \geq 1$ (Ji & Telgarsky, 2019, Lemma 5.4, first part, which does not depend on linear predictors). Overall, these dual variables match the usual interpretation in margin problems of corresponding to examples of high error, and also relate to the Lagrange multipliers used in the definition of KKT point.

On the topic of KKT points, further detail on the formalism is as follows. Firstly, (Lyu & Li, 2019) provided a definition for general $L$-homogeneous models, and the version here is equivalent for the simplified choice of 2-homogeneous models of the form $x \mapsto \sum_j a_j\sigma(v_j^\intercal x)$. Given

a KKT point $W$, the complementary slackness conditions imply $\min_i p_i(W) \geq 1$, whereby $\gamma(W) = \min_i p_i(W)/\|W\|^2 \geq 1/\|W\|^2$, justifying the choice of $1/\|W\|^2$ as the margin. Lastly, given any arbitrary $W$ (not necessarily a KKT point), the optimality conditions on $a_j$ and $v_j$ hold iff they hold for any rescaling $rW$ (since the term $r$ appears on both sides), and thus $r$ can be adjusted to make the complementary slackness conditions tight, justifying the definition of a KKT direction's margin.

To close, this section will collect various estimates of $\gamma_{\text{ntk}}$ and $\gamma_{\text{gl}}$. Firstly, both function classes are universal approximators, and thus the assumption can be made to work for any prediction problem with pure conditional probabilities (Ji et al., 2020). Next, as a warmup, note the following estimates of $\gamma_{\text{ntk}}$ and $\gamma_{\text{gl}}$, for linear predictors, with an added estimate of showing the value of working with both layers in the definition of $\gamma_{\text{ntk}}$.

**Proposition B.1.** . *Suppose the data distribution is almost surely linearly separable: there exists $\|\bar{u}\| = 1$ and $\widehat{\gamma} > 0$ with $yx^\top \bar{u} \geq \widehat{\gamma}$ almost surely.*

1. *Choosing $\theta(a, v) := \big(0, \text{sgn}(a)\bar{u}\big) \cdot \mathbb{1}[\|(a, v)\| \leq 2]$, then Assumption 1.2 holds with $\gamma_{\text{ntk}} \geq \frac{\widehat{\gamma}}{32}$.*

2. *Choosing $\theta(a, v) := \big(\text{sgn}(\bar{u}^\top v), 0\big) \cdot \mathbb{1}[\|(a, v)\| \leq 2]$, then Assumption 1.2 holds with $\gamma_{\text{ntk}} \geq \frac{\widehat{\gamma}}{16\sqrt{d}}$.*

3. *Choosing $\alpha = (1/2, -1/2)$ and $\beta = (\bar{u}, -\bar{u})$, then Assumption 1.1 holds with $\gamma_{\text{gl}} \geq \frac{\widehat{\gamma}}{2}$.*

*Proof.* The proof considers the three settings separately; in each, let $(x, y)$ be a random draw, which almost surely satisfies $\bar{u}^\top xy \geq \widehat{\gamma}$.

1. To start,

$$\mathbb{E}_{w \sim \mathcal{N}_\theta} \left\langle \theta(w), \hat{\partial}_w p(x, y; w) \right\rangle = \mathbb{E}_{(a,v) \sim \mathcal{N}_\theta} |a| \bar{u}^\top xy \sigma'(v^\top x) \mathbb{1}[\|(a, v)\| \leq 2]$$
$$\geq \widehat{\gamma} \mathbb{E}_{(a,v) \sim \mathcal{N}_\theta} |a| \sigma'(v^\top x) \mathbb{1}[\|(a, v)\| \leq 2].$$

To control the expectation, note that with probability at least $1/2$, then $1/4 \leq |a| \leq \sqrt{2}$, and thus by rotational invariance

$$\mathbb{E}_{(a,v) \sim \mathcal{N}_\theta} |a| \sigma'(v^\top x) \mathbb{1}[\|(a, v)\| \leq 2] \geq \frac{1}{8} \mathbb{E}_{(a,v) \sim \mathcal{N}_\theta} \sigma'(v^\top x) \mathbb{1}[\|v\| \leq \sqrt{2}]$$
$$\geq \frac{1}{8} \mathbb{E}_{(a,v) \sim \mathcal{N}_\theta} \sigma'(v_1) \mathbb{1}[\|v\| \leq \sqrt{2}]$$
$$\geq \frac{1}{32}.$$

2. For convenience, write $(a, v) = w$, whereby $w \sim \mathcal{N}_w$ means $a \sim \mathcal{N}_a$ and $v \sim \mathcal{N}_v$. With this out of the way, define orthonormal matrix $M \in \mathbb{R}^{d \times d}$ where the first column is $\bar{u}$, the second column is $(I - \bar{u}\bar{u}^\top)x/\|(I - \bar{u}\bar{u}^\top)x\|$, and the remaining columns are arbitrary so long as $M$ is orthonormal, and note that $Mu = e_1$ and $Mx = e_1\bar{u}^\top x + e_2 r_2$ where $r_2 := \sqrt{\|x\|^2 - (\bar{u}^\top x)^2}$. Then, using rotational invariance of the Gaussian,

$$\mathbb{E}_w \left\langle \theta(w), \hat{\partial}p(x, y; w) \right\rangle = y\mathbb{E}_{w=(a,v)}\text{sgn}(\bar{u}^\top v)\sigma(v^\top x)\mathbb{1}[\|w\| \leq 2]$$
$$= y\mathbb{E}_{\|(a, Mv)\| \leq 2}\alpha(Mv)\sigma(v^\top M^T x)$$
$$= \mathbb{E}_{\|(a,v)\| \leq 2}y\text{sgn}(v_1)\sigma(v_1\bar{u}^\top xy^2 + v_2 r_2)$$
$$= \mathbb{E}_{\|(a,v)\| \leq 2}y\text{sgn}(v_1)\sigma(y\text{sgn}(v_1)|v_1|\bar{u}^\top xy + v_2 r_2)$$
$$= \mathbb{E}_{\substack{\|(a,v)\| \leq 2 \\ y\text{sgn}(v_1)=1 \\ v_2 \geq 0}} \Big[\sigma(|v_1|\bar{u}^\top xy + v_2 r_2) - \sigma(-|v_1|\bar{u}^\top xy + v_2 r_2)$$
$$+ \sigma(|v_1|\bar{u}^\top xy - v_2 r_2) - \sigma(-|v_1|\bar{u}^\top xy - v_2 r_2)\Big].$$

Considering cases, the first ReLU argument is always positive, exactly one of the second and third is positive, and the fourth is negative, whereby

$$
\begin{aligned}
y\mathbb{E}_{\|(a,v)\|\leq 2}\alpha(v)\sigma(v^\mathsf{T}x) &= \mathbb{E}_{\substack{\|(a,v)\|\leq 2\\ y\mathrm{sgn}(v_1)=1\\ v_2\geq 0}}\left[|v_1|\bar{u}^\mathsf{T}xy + v_2 r_2 + |v_1|\bar{u}^\mathsf{T}xy - v_2 r_2\right]\\
&= 2\mathbb{E}_{\substack{\|(a,v)\|\leq 2\\ y\mathrm{sgn}(v_1)=1}}|v_1|\bar{u}^\mathsf{T}xy\\
&\geq 2\widehat{\gamma}\mathbb{E}_{\substack{\|v\|\leq 1\\ y\mathrm{sgn}(v_1)=1}}|v_1|\\
&= \widehat{\gamma}\Pr[\|(a,v)\|\leq 2]\mathbb{E}\left(|v_1|\,\big|\,\|(a,v)\|\leq 2\right),
\end{aligned}
$$

where $\Pr[\|(a,v)\|\leq 2]\geq 1/4$ since (for example) the $\chi^2$ random variables corresponding to $|a|^2$ and $\|v\|^2$ have median less than one, and the expectation term is at least $1/(4\sqrt{d})$ by standard Gaussian computations (Blum et al., 2017, Theorem 2.8).

3. It suffices to note that

$$
\begin{aligned}
2y\sum_{j=1}^{2}\alpha_j\sigma(\beta_j^\mathsf{T}x) &= y\sigma(\bar{u}^\mathsf{T}x) - y\sigma(-\bar{u}^\mathsf{T}x)\\
&= \mathbb{1}[y=1]\sigma(y\bar{u}^\mathsf{T}x) + \mathbb{1}[y=-1]\sigma(y\bar{u}^\mathsf{T}x)\\
&= y\bar{u}^\mathsf{T}x\\
&\geq \widehat{\gamma}.
\end{aligned}
$$

$\square$

Next, estimates for $\gamma_{\mathrm{gl}}$ and $\gamma_{\mathrm{ntk}}$ on 2-sparse parity were stated in the body in Proposition 1.3. The key is that $\gamma_{\mathrm{ntk}}$ scales with $1/d$ whereas $\gamma_{\mathrm{gl}}$ scales with $1/\sqrt{d}$, which suffices to yield the separations in Table 1. The bound on $\gamma_{\mathrm{ntk}}$ is also necessarily an upper bound, since otherwise it would be possible to beat the NTK lower bound (Wei et al., 2018).

*Proof of Proposition 1.3.* This proof shares ideas with (Wei et al., 2018; Ji & Telgarsky, 2020b), though with some adjustments to exactly fit the standard 2-sparse parity setting, and to shorten the proofs.

Without loss of generality, due to the symmetry of the data distribution about the origin, suppose $a = 1$ and $b = 2$, meaning for any $x \in H_d$, the correct label is $dx_1x_2$, the product of the first two coordinates. Both proofs will use the global margin construction (the parameters for $\gamma_{\mathrm{gl}}$), given as follows: $p(x,y;(\alpha,\beta)) = y\sum_{j=1}^{4}\alpha_j\sigma(\beta_j^\mathsf{T}x)$, where $\alpha = (1/4, -1/4, -1/4, 1/4)$ and

$$
\begin{aligned}
\beta_1 &:= \left(\frac{1}{\sqrt{2}}, \frac{1}{\sqrt{2}}, 0, \ldots, 0\right) \in \mathbb{R}^d,\\
\beta_2 &:= \left(\frac{1}{\sqrt{2}}, \frac{-1}{\sqrt{2}}, 0, \ldots, 0\right) \in \mathbb{R}^d,\\
\beta_3 &:= \left(\frac{-1}{\sqrt{2}}, \frac{1}{\sqrt{2}}, 0, \ldots, 0\right) \in \mathbb{R}^d,\\
\beta_4 &:= \left(\frac{-1}{\sqrt{2}}, \frac{-1}{\sqrt{2}}, 0, \ldots, 0\right) \in \mathbb{R}^d.
\end{aligned}
$$

Note moreover that for any $x \in H_d$, then $\beta_j^\mathsf{T}x > 0$ for exactly one $j$, which will be used for both $\gamma_{\mathrm{ntk}}$ and $\gamma_{\mathrm{gl}}$. The proof now splits into the two different settings, and will heavily use symmetry within $H_d$ and also within $(\alpha,\beta)$.

1. Consider the transport mapping

$$
\theta\left((a,v)\right) = \left(0, \frac{\mathrm{sgn}(a)}{2}\sum_{j=1}^{4}\beta_j\mathbb{1}[\beta_j^\mathsf{T}v \geq 0]\right);
$$

note that this satisfies the condition $\|\theta(w)\| \leq 1$ thanks to the factor $1/2$, since each $\beta_j$ gets a hemisphere, and $(\beta_1, \beta_4)$ together partition the sphere once, and $(\beta_2, \beta_3)$ similarly together partition the sphere once.

Now let any $x$ be given, which as above has label $y = x_1 x_2$. By rotational symmetry of the data and also the transport mapping, suppose suppose $\beta_1$ is the unique choice with $\beta_1^\mathsf{T} x > 0$, which implies $y = 1$, and also $\beta_2^\mathsf{T} x = 0 = \beta_3^\mathsf{T} x = 0$, however $\beta_4^\mathsf{T} x = -\beta_4^\mathsf{T} x$. Using these observations, and also rotational invariance of the Gaussian,

$$\mathbb{E}_{a,v}\left\langle \theta(a,v), \bar\partial p(x,y;w) \right\rangle$$

$$= \mathbb{E}_{a,v} \frac{|a|}{2} \sum_{j=1}^{4} \beta_j^\mathsf{T} x \mathbb{1}[\beta_j^\mathsf{T} v \geq 0] \cdot \mathbb{1}[v^\mathsf{T} x \geq 0]$$

$$= \beta_1^\mathsf{T} x \left( \mathbb{E}_a \frac{|a|}{2} \right) \cdot \left( \mathbb{E}_v \mathbb{1}[\beta_1^\mathsf{T} v \geq 0] \cdot \mathbb{1}[v^\mathsf{T} x \geq 0] - \mathbb{E}_v \mathbb{1}[-\beta_1^\mathsf{T} v \geq 0] \cdot \mathbb{1}[v^\mathsf{T} x \geq 0] \right).$$

Now consider $\mathbb{E}_v \mathbb{1}[\beta_1^\mathsf{T} v \geq 0] \cdot \mathbb{1}[v^\mathsf{T} x \geq 0]$. A standard Gaussian computation is to introduce a rotation matrix $M$ whose first column is $\beta_1$, whose second column is $(I - \beta_1\beta_1^\mathsf{T})x/\|(I - \beta_1\beta_1^\mathsf{T})x\|$, and the rest are orthogonal, which by rotational invariance and the calculation $\beta_1^\mathsf{T} x = \sqrt{2/d}$ gives

$$\mathbb{E}_v \mathbb{1}[\beta_1^\mathsf{T} v \geq 0] \cdot \mathbb{1}[v^\mathsf{T} x \geq 0] = \mathbb{E}_v \mathbb{1}[\beta_1^\mathsf{T} Mv \geq 0] \cdot \mathbb{1}[v^\mathsf{T} Mx \geq 0]$$

$$= \mathbb{E}_v \mathbb{1}[v_1 \geq 0] \cdot \mathbb{1}[v_1 \beta_1^\mathsf{T} x + v_2 \sqrt{1 - (\beta_1^\mathsf{T} x)^2}$$

$$= \mathbb{E}_v \mathbb{1}[v_1 \geq 0] \cdot \mathbb{1}[v_1 + v_2 \sqrt{d/2 - 1} \geq 0].$$

Performing a similar calculation for the other term (arising from $\beta_4^\mathsf{T} x$) and plugging all of this back in,

$$\mathbb{E}_{a,v}\left\langle \theta(a,v), \bar\partial p(x,y;w) \right\rangle$$

$$= \sqrt{\frac{2}{d}} \left( \mathbb{E}_a \frac{|a|}{2} \right) \cdot \mathbb{E}_v \mathbb{1}[v_1 \geq 0] \left( \mathbb{1}[v_1 + v_2 \sqrt{d/2 - 1} \geq 0] - \mathbb{1}[-v_1 + v_2 \sqrt{d/2 - 1} \geq 0] \right).$$

To finish, a few observations suffice. Whenever $v_1 \geq 0$ (which is enforced by the common first term), then $-v_1 + v_2\tau \leq v_1 + v_2\sqrt{d/2 - 1}$, so the first indicator is $1$ whenever the second indicator is $1$, thus their difference is nonnegative, and to lower bound the overall quantity, it suffices to asses the probability that $v_1 + v_2\sqrt{d/2 - 1} \geq 0$ whereas $-v_1 + v_2\sqrt{d/2 - 1} \leq 0$. To lower bound this event, it suffices to lower bound

$$\Pr[v_1 \geq 0 \,\wedge\, v_2 \geq 0 \,\wedge\, v_1 \geq v_2\sqrt{d/2 - 1}] \geq \Pr[v_1 \geq \sqrt{1/2}] \cdot \Pr[0 \leq v_2 \leq \sqrt{1/d}.$$

The first term is at least $1/5$, and the second can be calculated via brute force:

$$\Pr[v_2 \geq 1/\sqrt{d}] = \frac{1}{\sqrt{2\pi}} \int_0^{1/\sqrt{d}} \exp(-x^2)\,\mathrm{d}x \geq \frac{1}{\sqrt{2\pi}} \int_0^{1/\sqrt{d}} \exp(-1/d)\,\mathrm{d}x \geq \frac{1}{\sqrt{2\pi}} \left( \frac{1}{\sqrt{d}} \right) \frac{1}{e},$$

which completes the proof after similarly using $\mathbb{E}_a |a| \geq 1$, and simplifying the various constants.

2. Let any $x \in H_d$ be given, and as above note that $\beta_j^\mathsf{T} x > 0$ for exactly one $j$. By symmetry, suppose it is $\beta_1$, whereby $y = x_1 x_2 = 1$, and

$$\gamma_{\text{gl}} \geq p(x,y;(\alpha,\beta)) = y \sum_j \alpha_j \sigma(\beta_j^\mathsf{T} x) = |\alpha_1| \cdot \beta_1^\mathsf{T} x = \frac{1}{4}\left( \frac{2}{\sqrt{2d}} \right) = \frac{1}{\sqrt{8d}}.$$

$\square$

To close, consider the *k-sparse parity problem*, the natural $k$-bit analog of the 2-sparse parity problem: now the target label depends on the product of $k$ unknown input bits, but otherwise the problem is the same, meaning the data distribution is again supported on $H_d := \{\pm 1/\sqrt{d}\}^d$, and only the support of the distribution is used in the margin analysis.

**Proposition B.2.** *Let $k \geq 4$ be an even integer, and consider any k-sparse parity data distribution: inputs are supported on $H_d := \{\pm 1/\sqrt{d}\}^d$ (as in Proposition 1.3), and for any $x \in H_d$, the label is the product of k fixed coordinates: $y := d^{k/2} \prod_{i \in S} x_i$ with $|S| = k$. Then Assumption 1.1 holds with $\gamma_{\mathrm{gl}} \geq \frac{1}{2k\sqrt{d}}$.*

*Proof.* Let $\mathcal{P}(S)$ range over the $2^k$ possible vectors which are $\pm 1/\sqrt{k}$ on elements of $S$, and $0$ otherwise, whereby $v \in \mathcal{P}(S)$ has $\|v\| = 1$. Moreover, for convenience, define a shorthand $\mathrm{sgn}(x) = \mathrm{sgn}(\prod_{i \in S} x_i)$. With this in hand, define a target mapping

$$h(x) := \frac{(-1)^{k/2-1}}{2^k} \sum_{v \in \mathcal{P}(S)} \mathrm{sgn}(v) \sigma(v^\mathsf{T} x),$$

which is of the desired form for Assumption 1.1 with $\alpha_v = \mathrm{sgn}(v)(-1)^{k/2-1}/2^k$ and $\beta_v = v$, and moreover $\sum_v |\alpha_v| = 1$ and $\|\beta_v\|_2 = 1$.

Now let any $x \in H_d$ with corresponding label $y = \mathrm{sgn}(x)$ be given. To develop a first simplification of the margin $yg(x)$, let $\mathcal{P}(S)[x; j]$ denote $v \in \mathcal{P}(S)$ where the signs of $x$ and $v$ have $j$ disagreements within the support of $S$, whereby $|\mathcal{P}(S)[x; j]| = \binom{k}{j}$ and

$$
\begin{aligned}
yg(x) &= \mathrm{sgn}(x) \sum_{v \in \mathcal{P}(S)} \frac{(-1)^{k/2-1}\mathrm{sgn}(v)}{2^k} \sigma(v^\mathsf{T} x) \\
&= (-1)^{k/2-1} \sum_{j=0}^{k} \sum_{v \in \mathcal{P}(S)[x;j]} \frac{\mathrm{sgn}(v)\mathrm{sgn}(x)}{2^k} \sigma\left(\frac{k-2j}{\sqrt{kd}}\right) \\
&= (-1)^{k/2-1} \sum_{j=0}^{k/2} \sum_{v \in \mathcal{P}(S)[x;j]} \frac{(-1)^j}{2^k} \left(\frac{k-2j}{\sqrt{kd}}\right) \\
&= \frac{(-1)^{k/2-1}}{2^k \sqrt{kd}} \sum_{j=0}^{k/2} (-1)^j \binom{k}{j}(k-2j).
\end{aligned}
$$

The inner sum can now be handled via a few binomial tricks from (Graham et al., 1994, Chapter 5):

$$\sum_{j=0}^{k/2}(-1)^j\binom{k}{j}(k-2j)$$

$$= k + \sum_{j=1}^{k/2}(-1)^j\binom{k}{j}(k-2j)$$

$$= k + k\sum_{j=1}^{k/2}(-1)^j\binom{k-1}{j-1}\frac{k-2j}{j} \qquad \text{(Graham et al., 1994, eq. (5.6))}$$

$$= k + k\sum_{j=1}^{k/2}(-1)^{j-1}\binom{k-1}{j-1}\frac{2j-k}{j}$$

$$= k + k\sum_{j=1}^{k/2}\binom{j-1-(k-1)-1}{j-1}\frac{2j-k}{j} \qquad \text{(Graham et al., 1994, eq. (5.14))}$$

$$= k + k\sum_{j=1}^{k/2}\left[\binom{j-k-1}{j-1}\frac{j-k}{j} + \binom{j-k-1}{j-1}\right]$$

$$= k\sum_{j=0}^{k/2}\binom{j-k}{j} + k\sum_{j=0}^{k/2-1}\binom{j-k}{j}$$

$$= k\left[\binom{k/2+1-k}{k/2} + \binom{k/2-k}{k/2-1}\right] \qquad \text{(Graham et al., 1994, eq. (5.9))}$$

$$= k\left[(-1)^{k/2}\binom{k-2}{k/2} + (-1)^{k/2-1}\binom{k-2}{k/2-1}\right]. \qquad \text{(Graham et al., 1994, eq. (5.14))}$$

As an elementary simplification,

$$\binom{k-2}{k/2-1} - \binom{k-2}{k/2} = \frac{(k-2)!}{(k/2-1)!(k/2-1)!} - \frac{(k-2)!}{(k/2-2)!(k/2)!}$$

$$= \frac{(k-3)!}{(k/2-2)!(k/2-1)!}\left[\frac{k-2}{k/2-1} - \frac{k-2}{k/2}\right]$$

$$= \binom{k-3}{k/2-1}\frac{4}{k},$$

which combines with the preceding to give

$$\sum_{j=0}^{k/2}(-1)^j\binom{k}{j}(k-2j) = k\left[(-1)^{k/2}\binom{k-2}{k/2} + (-1)^{k/2-1}\binom{k-2}{k/2-1}\right]$$

$$= k(-1)^{k/2-1}\binom{k-3}{k/2-1}\frac{4}{k},$$

which after combining with the original simplification gives

$$yg(x) = \frac{(-1)^{k/2-1}}{2^k\sqrt{kd}}\sum_{j=0}^{k/2}(-1)^j\binom{k}{j}(k-2j)$$

$$= \frac{(-1)^{k/2-1}}{2^k\sqrt{kd}}\left[k(-1)^{k/2-1}\binom{k-3}{k/2-1}\frac{4}{k}\right]$$

$$= \frac{4}{2^k\sqrt{kd}}\binom{k-3}{k/2-1}.$$

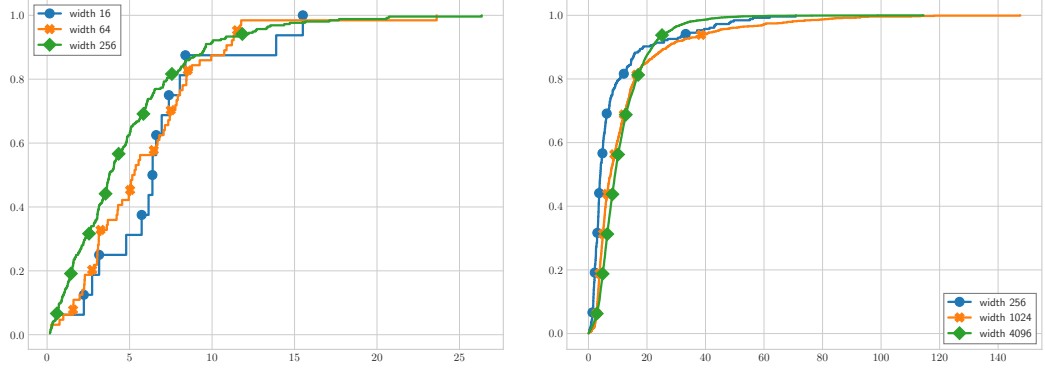

(a) Norms for 2-sparse parity, $m \in \{16, 64, 256\}$.

(b) Norms for `mnist`, $m \in \{256, 1024, 4096\}$.

Figure 3: CDFs of norm growth on 2-sparse parity and `mnist` digits 3 vs 5, with three choices of width, and all other experimental details as in Figure 1, Figure 2, and Appendix B.2. Here, for each width $m$, the norm growth of node $j$ is interpreted as $\sqrt{m}\|v_j(t) - v_j(0)\|$; the $\sqrt{m}$ factor is due to the gradient of $v_j$ scaling with $a_j$, which initially has magnitude roughly $1/\sqrt{m}$, and results in overlapping CDFs. One measure of exiting the NTK, though a bit weak, is that most rescaled norms are far beyond 1; an experiment to finer accuracy could be an interesting direction for future work.

To close, for the final estimate, if $k = 4$ then

$$\frac{4}{2^k\sqrt{kd}}\binom{k-3}{k/2-1} = \frac{4}{2^4\sqrt{4d}}\binom{1}{1} = \frac{1}{2k\sqrt{d}},$$

and otherwise, if $k > 4$, note firstly via standard lower bounds on the central binomial coefficient that

$$\binom{k-3}{k/2-1} = \frac{k-3}{k/2-2}\binom{k-2}{k/2-1} \geq \frac{k-3}{k/2-2}\left(\frac{2^{k-2}}{\sqrt{k-1}}\right) \geq 2\left(\frac{2^{k-2}}{\sqrt{k-1}}\right),$$

and thus

$$\frac{4}{2^k\sqrt{kd}}\binom{k-3}{k/2-1} \geq \frac{2}{(k-1)\sqrt{kd}} \geq \frac{2}{k\sqrt{d}}.$$

$\square$

### B.2 Experimental details

This brief section summarizes various choices used in the experiments behind Figure 1 and Figure 2, and provides an additional companion figure Figure 3.

The `mnist` data was limited to classes 3 and 5 to give a binary classification problem which is not linearly separable, and otherwise unmodified. The 2-sparse parity data was uniform over $H_d$, the corners of the rescaled hypercube as defined in Proposition 1.3, and further described in Figure 1 (e.g., $n = 64$ samples and $d = 20$ dimensions).

To simulate gradient flow, full-batch gradient descent was used together with the logistic loss. Initially the step size was 0.01, but eventually the `mnist` plots switched to step size 1, which did not lead to any discernible change. All experiments were run until the empirical logistic risk was approximately $1/n$.

A companion figure to Figure 2 from the paper body is to plot the CDFs of norm changes of the inner layer; this is presented here in Figure 3, and as detailed in the caption, also indicates an exit to the NTK, though it is unclear if it is quite to the significant level $\|W_t - W_0\| = \mathcal{O}(\sqrt{m})$ allowed by the theoretical guarantees.

The first concentration inequalities are purely about the initialization.

**Lemma B.3.** *Suppose $a \sim \mathcal{N}_m/\sqrt{m}$ and $V \sim \mathcal{N}_{m \times d}/\sqrt{d}$.*

1. *With probability at least $1 - \delta$, then $\|a\| \leq 1 + \sqrt{2\ln(1/\delta)/m}$; similarly, with probability at least $1 - \delta$, then $\|V\| \leq \sqrt{m} + \sqrt{2\ln(1/\delta)/d}$.*

2. *Let examples $(x_1, \ldots, x_n)$ be given with $\|x_i\| \leq 1$. With probability at least $1 - 4\delta$,*

$$\max_i \left| \sum_j a_j \sigma(v_j^\mathsf{T} x_i) \right| \leq 4\ln(n/\delta).$$

*Proof.*   1. Rewrite $\tilde{a} := a\sqrt{m}$, so that $\tilde{a} \sim \mathcal{N}_m$. Since $\tilde{a} \mapsto \|\tilde{a}\|/\sqrt{m} = \|a\|$ is $(1/\sqrt{m})$-Lipschitz, then by Gaussian concentration, (Wainwright, 2019, Theorem 2.26),

$$\begin{aligned} \|a\| &= \|\tilde{a}\|/\sqrt{m} \\ &\leq \mathbb{E}\|\tilde{a}\|/\sqrt{m} + \sqrt{2\ln(1/\delta)/m} \\ &\leq \sqrt{\mathbb{E}\|\tilde{a}\|^2}/\sqrt{m} + \sqrt{2\ln(1/\delta)/m} \\ &= 1 + \sqrt{2\ln(1/\delta)/m}. \end{aligned}$$

Similarly for $V$, defining $\tilde{V} := V\sqrt{d}$ whereby $\tilde{V} \sim \mathcal{N}_{m \times d}$, Gaussian concentration grants

$$\|V\| = \|\tilde{V}\|/\sqrt{d} \leq \sqrt{m} + \sqrt{2\ln(1/\delta)/d}.$$

2. Fix any example $x_i$, and constants $\epsilon_1 > 0$ and $\epsilon_2 > 0$ to be optimized at the end of the proof, and define $d_i := d/\|x_i\|^2$ for convenience. By rotational invariance of Gaussians and since $x_i$ is fixed, then $\sigma(Vx_i)$ is equivalent in distribution to $\|x_i\|\sigma(g)/\sqrt{d} = \sigma(g)/\sqrt{d_i}$ where $g \sim \mathcal{N}_m$. Meanwhile, $g \mapsto \|\sigma(g)\|/\sqrt{d_i}$ is $(1/\sqrt{d_i})$-Lipschitz with $\mathbb{E}\|\sigma(g)\| \leq \sqrt{m}$, and so, by Gaussian concentration (Wainwright, 2019, Theorem 2.26),

$$\Pr[\|\sigma(Vx_i)\| \geq \epsilon_1 + \sqrt{m}] = \Pr[\|\sigma(g)\|/\sqrt{d_i} \geq \epsilon_1 + \sqrt{m}] \leq \exp\left(\frac{-d_i\epsilon_1^2}{2}\right).$$

Next consider the original expression $a^\mathsf{T}\sigma(Vx_i)$. To simplify handling of the $1/m$ variance of the coordinates of $a$, define another Gaussian $h := a\sqrt{m}$, and a new constant $c_i := md_i$ for convenience, whereby $a^\mathsf{T}\sigma(V_i)$ is equivalent in distribution to equivalent in distribution to $h^\mathsf{T}\sigma(g)/\sqrt{c_i}$ since $a$ and $V$ are independent (and thus $h$ and $V$ are independent). Conditioned on $g$, since $\mathbb{E}h = 0$, then $\mathbb{E}[h^\mathsf{T}\sigma(g)|g] = 0$. As such, applying Gaussian concentration to this conditioned random variable, since $h \mapsto h^\mathsf{T}\sigma(g)/\sqrt{c_i}$ is $(\|\sigma(g)\|/\sqrt{c_i})$-Lipschitz, then

$$\Pr[h^\mathsf{T}\sigma(g)/\sqrt{c_i} \geq \epsilon_2 \mid g] \leq \exp\left(\frac{-c_i\epsilon_2^2}{2\|\sigma(g)\|^2}\right).$$

Returning to the original expression, it can now be controlled via the two preceding bounds, conditioning, and the tower property of conditional expectation:

$\Pr[h^\intercal\sigma(g)/\sqrt{c_i} \geq \epsilon_2]$

$$\leq \Pr\left[h^\intercal\sigma(g)/\sqrt{c_i} \geq \epsilon_2 \mid \|\sigma(g)\|/\sqrt{d_i} \leq \epsilon_1 + \sqrt{m}\right] \cdot \Pr\left[\|\sigma(g)\|/\sqrt{d_i} \leq \epsilon_1 + \sqrt{m}\right]$$

$$+ \Pr\left[h^\intercal\sigma(g)/\sqrt{c_i} \geq \epsilon_2 \mid \|\sigma(g)\|/\sqrt{d_i} > \epsilon_1 + \sqrt{m}\right] \cdot \Pr\left[\|\sigma(g)\|/\sqrt{d_i} > \epsilon_1 + \sqrt{m}\right]$$

$$= \mathbb{E}\left[\Pr[h^\intercal\sigma(g)/\sqrt{c_i} \geq \epsilon_2 \mid g] \;\Big|\; \|\sigma(g)\|/\sqrt{d_i} \leq \epsilon_1 + \sqrt{m}\right] \Pr[\|\sigma(g)\|/\sqrt{d_i} \leq \epsilon_1 + \sqrt{m}]$$

$$+ \Pr[h^\intercal\sigma(g)/\sqrt{c_i} \geq \epsilon_2 \mid \|\sigma(g)\|/\sqrt{d_i} > \epsilon_1 + \sqrt{m}]\Pr[\|\sigma(g)\|/\sqrt{d_i} > \epsilon_1 + \sqrt{m}]$$

$$\leq \mathbb{E}\left[\exp\left(\frac{-c_i\epsilon_2^2}{2\|\sigma(g)\|^2}\right) \;\Big|\; \|\sigma(g)\|/\sqrt{d_i} \leq \epsilon_1 + \sqrt{m}\right] + \exp\left(-d_i\epsilon_1^2/2\right)$$

$$\leq \exp\left(\frac{-c_i\epsilon_2^2}{4d_i\epsilon_1^2 + 4d_i m}\right) + \exp\left(-d_i\epsilon_1^2/2\right).$$

As such, choosing $\epsilon_2 := 4\ln(n/\delta)\sqrt{md_i/c_i} = 4\ln(n/\delta)$ and $\epsilon_1 := \sqrt{2\ln(n/\delta)/d_i}$ gives

$$\Pr[a^\intercal\sigma(Vx_i) \geq \epsilon_2] = \Pr[h^\intercal\sigma(g)/\sqrt{c_i} \geq \epsilon_2] \leq \frac{\delta}{n} + \frac{\delta}{n},$$

which is a sub-exponential concentration bound. Union bounding over the reverse inequality and over all $n$ examples and using $\max_i \|x_i\| \leq 1$ gives the final bound.

$\square$

Next comes a key tool in all the proofs using $\gamma_{\text{ntk}}$: guarantees that the infinite-width margin assumptions imply the existence of good finite-width networks.

**Lemma B.4.** *Suppose the data distribution satisfies Assumption 1.2 with corresponding $\theta : \mathbb{R}^{d+1} \to \mathbb{R}^{d+1}$ and $\gamma_{\text{ntk}} > 0$, and let $((x_i, y_i))_{i=1}^n$ be an iid draw.*

1. *With probability at least $1 - \delta$ over the draw of $(w_j)_{j=1}^m$, defining $\overline{\theta}_j := \theta(w_j)/\sqrt{m}$, then*

$$\min_i \sum_j \left\langle \overline{\theta}_j, \hat{\partial}p_i(w_j) \right\rangle \geq \gamma_{\text{ntk}}\sqrt{m} - \sqrt{32\ln(n/\delta)}.$$

2. *With probability at least $1 - 7\delta$ over the draw of $W$ with rows $(w_j)_{j=1}^m$ with $m \geq 2\ln(1/\delta)$, defining rows $\overline{\theta}_j := \theta(w_j)/\sqrt{m}$ of $\overline{\theta} \in \mathbb{R}^{m \times (d+1)}$, then for any $W'$ and any $R \geq \|W - W'\|$ and any $r_\theta \geq 0$ and $r_w \geq 0$,*

$$\left\langle r_\theta\overline{\theta} + r_w W, \hat{\partial}p_i(W') \right\rangle - r_w p_i(W') \geq \gamma_{\text{ntk}} r_\theta \sqrt{m} - r_\theta\left[\sqrt{32\ln(n/\delta)} + 8R + 4\right]$$
$$- r_w\left[4\ln(n/\delta) + 2R + 2R\sqrt{m} + 4\sqrt{m}\right],$$

*and moreover, writing $W = (a, V)$, then $\|a\| \leq 2$ and $\|V\| \leq 2\sqrt{m}$. For the particular choice $r_\theta := R/8$ and $r_w = 1$, if $R \geq 8$ and $m \geq (64\ln(n/\delta)/\gamma_{\text{ntk}})^2$, then*

$$\left\langle r_\theta\overline{\theta} + W, \hat{\partial}p_i(W') \right\rangle - p_i(W') \geq \frac{\gamma_{\text{ntk}} r_\theta \sqrt{m}}{2} - 160 r_\theta^2.$$

*Proof.* 1. Fix any example $(x_i, y_i)$, and define

$$\mu := \mathbb{E}_w \left\langle \theta(w), \hat{\partial}p_i(w) \right\rangle,$$

where $\mu \geq \gamma_{\text{ntk}}$ by assumption. By the various conditions on $\theta$, it holds for any $(a, v) := w \in \mathbb{R}^{d+1}$ and corresponding $(\overline{a}, \overline{v}) := \theta(w) \in \mathbb{R}^{d+1}$ that

$$\left|\left\langle \theta(w), \hat{\partial}p_i(w) \right\rangle\right| \leq \left|\overline{a}\sigma(v^\intercal x_i)\right| + \left|\left\langle \overline{v}, ax_i\sigma'(v^\intercal x_i) \right\rangle\right|$$
$$\leq |\overline{a}| \cdot \mathbb{1}[\|v\| \leq 2] \cdot \|v\| \cdot \|x_i\| + \|\overline{v}\| \cdot |a| \cdot \mathbb{1}[|a| \leq 2] \cdot \|x_i\|$$
$$\leq 4.$$

and therefore, by Hoeffding's inequality, with probability at least $1 - \delta/n$ over the draw of $m$ iid copies of this random variable,

$$\sum_j \left\langle \theta(w_j), \hat{\partial} p_i(w_j) \right\rangle \geq m\mu - \sqrt{32m\ln(n/\delta)} \geq m\gamma_{\text{ntk}} - \sqrt{32m\ln(n/\delta)},$$

which gives the desired bound after dividing by $\sqrt{m}$, recalling $\overline{\theta}_j := \theta(w_j)/\sqrt{m}$, and union bounding over all $n$ examples.

2. First, suppose with probability at least $1 - 7\delta$ that the consequences of Lemma B.3 and the preceding part of the current lemma hold, whereby simultaneously $\|a\| \leq 2$, and $\|V\| \leq 2\sqrt{m}$, and

$$\min_i p_i(W) \geq -4\ln(n/\delta), \qquad \text{and} \qquad \min_i \sum_j \left\langle \overline{\theta}_j, \hat{\partial} p_i(w_j) \right\rangle \geq \gamma_{\text{ntk}}\sqrt{m} - \sqrt{32\ln(n/\delta)}.$$

The remainder of the proof proceeds by separately lower bounding the two right hand terms in

$$\left\langle r_\theta \overline{\theta} + r_w W, \hat{\partial} p_i(W') \right\rangle - r_w p_i(W') = r_\theta \left[ \left\langle \overline{\theta}, \hat{\partial} p_i(W) \right\rangle + \left\langle \overline{\theta}, \hat{\partial} p_i(W') - \hat{\partial} p_i(W) \right\rangle \right]$$
$$+ r_w \left[ \left\langle W, \hat{\partial} p_i(W') \right\rangle - r_w p_i(W') \right].$$

For the first term, writing $(\overline{a}, \overline{\mathcal{V}}) = \overline{\theta}$ and noting $\|\overline{a}\| \leq 2$ and $\|\overline{\mathcal{V}}\| \leq 2$, then for any $W' = (a', V')$,

$$\left| \left\langle \overline{\theta}, \hat{\partial} p_i(W') - \hat{\partial} p_i(W) \right\rangle \right| \leq \left| \sum_j \overline{a}_j \left( \sigma(x_i^\mathsf{T} v_j') - \sigma(v_j^\mathsf{T} x_i) \right) \right|$$

$$+ \left| \sum_j x_i^\mathsf{T} \overline{v}_j \left( a_j' \sigma'(x^\mathsf{T} v_j') - a_j \sigma'(x^\mathsf{T} v_j) \right) \right|$$

$$\leq \sqrt{\sum_j \overline{a}_j^2} \sqrt{\sum_j \left( \sigma(x_i^\mathsf{T} v_j') - \sigma(v_j^\mathsf{T} x_i) \right)^2}$$

$$+ \sum_j |x_i^\mathsf{T} \overline{v}_j| \cdot \left| a_j' \sigma'(x^\mathsf{T} v_j') - a_j \sigma'(x^\mathsf{T} v_j') \right|$$

$$+ \sum_j |x_i^\mathsf{T} \overline{v}_j| \cdot \left| a_j \sigma'(x^\mathsf{T} v_j') - a_j \sigma'(x^\mathsf{T} v_j') \right|$$

$$\leq \|\overline{a}\| \cdot \|V' - V\| + \|a' - a\| \cdot \|\overline{\mathcal{V}}\| + \|a\| \cdot \|\overline{\mathcal{V}}\|$$
$$\leq 4R + 4.$$

For the second term,

$$\left| \left\langle W, \hat{\partial} p_i(W') \right\rangle - p_i(W') \right| = \left| \left\langle a, \hat{\partial}_a p_i(W') \right\rangle + \left\langle V, \hat{\partial}_V p_i(W') \right\rangle - \left\langle V', \hat{\partial}_V p_i(W') \right\rangle \right|$$

$$\leq \left| \sum_j a_j \sigma(x_i^\mathsf{T} v_j') \right| + \left| \sum_j a_j' \left\langle v_j - v_j', x_i \right\rangle \sigma'(x_i^\mathsf{T} v_j) \right|$$

$$\leq \left| p_i(w) + y_i \sum_j a_j \left( \sigma(x_i^\mathsf{T} v_j') - \sigma(x_i^\mathsf{T} v_j) \right) \right| + \sum_j \left| a_j' \right| \cdot \|v_j - v_j'\|$$

$$\leq 4\ln(n\delta) + \|a\| \cdot \|V - V'\| + \|a' - a + a\| \cdot \|V - V'\|$$
$$\leq 4\ln(n\delta) + 4R + R^2.$$

Multiplying through by $r_\theta$ and $r$ and combining these inequalities gives, for every $i$,

$$\left\langle r_\theta \overline{\theta} + r_w W, \hat{\partial} p_i(W') \right\rangle - r_w p_i(W') \geq \gamma_{\text{ntk}} r_\theta \sqrt{m} - r_\theta \left[ \sqrt{32\ln(n/\delta)} + 4R + 4 \right]$$
$$- r_w \left[ 4\ln(n/\delta) + 4R + R^2 \right],$$

which establishes the first inequality. For the particular choice $r_\theta := R/8$ with $R \geq 8$ and $r_w = 1$, and using $m \geq (64 \ln(n/\delta)/\gamma_{\mathrm{ntk}})^2$, the preceding bound simplifies to

$$
\left\langle r_\theta \overline{\theta} + r_w W, \hat{\partial} p_i(W') \right\rangle - r_w p_i(W') \geq \gamma_{\mathrm{ntk}} r_\theta \sqrt{m} - r_\theta \left[ \frac{\gamma_{\mathrm{ntk}} \sqrt{m}}{8} + 32 r_\theta + 32 r_\theta \right]
$$

$$
- \left[ \frac{\gamma_{\mathrm{ntk}} \sqrt{m}}{16} + 32 r_\theta + 64 r_\theta^2 \right]
$$

$$
\geq \frac{\gamma_{\mathrm{ntk}} r_\theta \sqrt{m}}{2} - 160 r_\theta^2.
$$

$\square$

## B.4   BASIC PROPERTIES OF $L$-HOMOGENEOUS PREDICTORS

This subsection collects a few properties of arbitrary $L$-homogeneous predictors in a setup more general than the rest of the work, and used in all large margin calculations. Specifically, suppose general parameters $u_t$ with some unspecified initial condition $u_0$, and thereafter given by the differential equation

$$
\dot{u}_t = -\overline{\partial}_u \widehat{\mathcal{R}}(p(u_t)), \tag{B.1}
$$

where

$$
\begin{aligned}
p(u) &:= (p_1(u), \ldots, p_n(u)) \in \mathbb{R}^n, \\
p_i(u) &:= y_i F(x_i; u), \\
F(x_i; cu) &= c^L F(x_i; u) && \forall c \geq 0.
\end{aligned}
$$

The first property is that norms increase once there is a positive margin.

**Lemma B.5** (Restatement of (Lyu & Li, 2019, Lemma B.1)). *Suppose the setting of eq. (B.1) and also $\ell \in \{\ell_{\exp}, \ell_{\log}\}$. If $\widehat{\mathcal{R}}(u_\tau) < \ell(0)/n$, then, for every $t \geq \tau$,*

$$
\frac{\mathrm{d}}{\mathrm{d}t} \|u_t\| > 0 \qquad \text{and} \qquad \langle u_t, \dot{u}_t \rangle > 0,
$$

*and moreover $\lim_t \|u_t\| = \infty$.*

*Proof.* Since $\widehat{\mathcal{R}}$ is nonincreasing during gradient flow, it suffices to consider any $u_s$ with $\widehat{\mathcal{R}}(u_s) < \ell(0)/n$. To apply (Lyu & Li, 2019, Lemma B.1), first note that both the exponential and logistic losses can be handled, e.g., via the discussion of the assumptions at the beginning of (Lyu & Li, 2019, Appendix A.1). Next, the statement of that lemma is

$$
\frac{\mathrm{d}}{\mathrm{d}s} \ln \|u_s\| > 0,
$$

but note that $\|u_s\| > 0$ (otherwise $\widehat{\mathcal{R}}(u_s) < \ell(0)/n$ is impossible), and also that

$$
\frac{\mathrm{d}}{\mathrm{d}s} \|u_s\| = \frac{\langle u_s, \dot{u}_s \rangle}{\|u_s\|}, \quad \text{and} \quad \frac{\mathrm{d}}{\mathrm{d}s} \ln \|u_s\| = \frac{\langle u_s, \dot{u}_s \rangle}{\|u_s\|^2},
$$

which together with $(\mathrm{d}/\mathrm{d}s) \ln \|u_s\| > 0$ from (Lyu & Li, 2019, Lemma B.1) imply the main part of the statement; all that remains to show is $\|u_s\| \to \infty$, but this is given by (Lyu & Li, 2019, Lemma B.6). $\square$

Next, even without the assumption $\widehat{\mathcal{R}}(u_s) < \ell(0)/n$ (which at a minimum requires a two-phase proof, and certain other annoyances), note that once $\|u_s\|$ is large, then the gradient can be related to margins, even if they are negative, which will be useful in circumventing the need for dual convergence and other assumptions present in prior work (e.g., as in (Chizat & Bach, 2020)). We note that while the closest inequalities in the literature require the condition $\widehat{\mathcal{R}}(u_s) < \ell(0)/n$ (Ji & Telgarsky, 2020a, Lemma C.5), those results aim for a more stringent goal, replacing $n$ in the bound below with $\ln(n)$; this simpler goal is sufficient in the present work.

**Lemma B.6** (See also (Ji & Telgarsky, 2020a, Proof of Lemma C.5)). *Suppose the setting of eq. (B.1) and also $\ell \in \{\ell_{\exp}, \ell_{\log}\}$. Then, for any $u$ and any $((x_i, y_i))_{i=1}^n$ (and corresponding $\widehat{\mathcal{R}}$),*

$$\frac{\langle u, -n\bar{\partial}_u \widehat{\mathcal{R}}(u) \rangle}{L\|u\|^L} \leq \mathcal{Q}\left[\mathring{\gamma}(u) + \frac{n}{\|u\|^L}\right] \leq \mathcal{Q}\left[\gamma(u) + \frac{n}{\|u\|^L}\right].$$

*Proof.* Define $v := p(u)$ for convenience, as well as $\pi(v) = \ell^{-1}(\sum_i \ell(v_i)) = \widetilde{\gamma}(u)$, whereby $q = \nabla_p \pi(v)$, and $\pi$ is (unconditionally) concave (Ji & Telgarsky, 2020a, Lemma C.8). Combining these facts,

$$\left\langle u, -n\bar{\partial}_u \widehat{\mathcal{R}}(u) \right\rangle = \sum_i -\ell'(v_i) \left\langle u, \bar{\partial}_u p_i(u) \right\rangle = LQ \sum_i q_i v_i = LQ \left\langle \nabla_v \pi(v), v \right\rangle$$

$$= LQ \left\langle \nabla_v \pi(v), v - 0 \right\rangle \leq LQ \left[\pi(v) - \pi(0)\right].$$

Simplifying $-\pi(0)$ now proceeds separately for $\ell_{\exp}$ and $\ell_{\log}$: for $\ell_{\exp}$, then

$$-\pi(0) = \ln(\sum_i \exp(0)) = \ln(n),$$

whereas for $\ell_{\log}$, then $\ell_{\log}^{-1}(r) = -\ln(e^r - 1)$, thus

$$-\pi(0) = \ln\left(\exp\left(\sum_i \ln(1 + \exp(-0))\right) - 1\right) = \ln\left(\exp\left(n \ln 2\right) - 1\right) = \ln\left(2^n - 1\right) \leq n \ln 2.$$

As such, in either case,

$$\left\langle u, -n\bar{\partial}_u \widehat{\mathcal{R}}(u) \right\rangle \leq LQ \left[\pi(v) - \pi(0)\right] \leq LQ \left[\widetilde{\gamma} + n\right].$$

Next, since $\ell$ is strictly decreasing in both cases, then $\ell^{-1}$ is strictly decreasing as well, whereby letting $k$ denote the index of any example with $v_k = \min_i v_i$, then additionally using the positivity of $\ell$ gives

$$\widetilde{\gamma} = \ell^{-1}\left(\sum_i \ell(v_i)\right) \leq \ell^{-1}\left(\ell(v_k)\right) = v_k = \gamma(u)\|u\|^L.$$

Combining these inequalities and dividing by $L\|u\|^L$ gives the desired bounds. $\square$

Lastly, a key abstract potential function lemma: this potential function is a proxy for mass accumulating on certain weights with good margin, and once it satisfies a few conditions, large margins are implied directly. This is the second component needed to remove dual convergence from (Chizat & Bach, 2020).

**Lemma B.7.** *Suppose the setting of eq. (B.1) with $L = 2$, and $\ell \in \{\ell_{\exp}, \ell_{\log}\}$. Then, unconditionally, $\lim_t \int_0^t \mathcal{Q}_s \, \mathrm{d}s = \infty$. Moreover, if there exists a constant $\widehat{\gamma} > 0$, a time $\tau$, and a potential function $\Phi(u)$ so that $\Phi(u_\tau) > -\infty$, and for all $t \geq \tau$,*

$$\Phi(u) \leq \frac{1}{L} \ln \|u\| \qquad \text{and} \qquad \frac{\mathrm{d}}{\mathrm{d}t} \Phi(u) \geq \frac{1}{n} \mathcal{Q}(u)\widehat{\gamma},$$

*then it follows that $\widehat{\mathcal{R}}(u) \to 0$, and $\|u\| \to \infty$, and $\liminf_t \gamma(u_t) \geq \widehat{\gamma}$.*

*Proof.* The unconditional claim $\int_0^t \mathcal{Q}_s \, \mathrm{d}s \to \infty$ is shown by considering two cases: either $\inf_s \widehat{\mathcal{R}}(u_s) = 0$, or $\widehat{\mathcal{R}}(u_s) > 0$ (the case $\inf_s \widehat{\mathcal{R}}(u_s) < 0$ is not possible since $\ell$ is nonnegative).

1. First suppose $\inf_s \widehat{\mathcal{R}}(u_s) > 0$. For both losses, it will be argued that $\inf_s \mathcal{Q}_s > 0$, whereby $\int_0^\infty \mathcal{Q}_s \, \mathrm{d}s \geq \int_0^\infty \inf_r \mathcal{Q}_r \, \mathrm{d}s = \infty$. In the case of $\ell_{\exp}$, then $\widehat{\mathcal{R}}(u_s) = \frac{1}{n}\mathcal{Q}_s$, and $\inf_s \mathcal{Q}_s > 0$ directly. In the case of $\ell_{\log}$, note $\ell_{\log}^{-1}(r) = -\ln(e^r - 1)$ and $\ell'(z) = -(1 + e^z)^{-1}$, whereby

$$(\ell'_{\log} \circ \ell_{\log}^{-1})(r) = \frac{-1}{1 + (e^r - 1)^{-1}} = \frac{1 - e^r}{e^r - 1 + 1} = \exp(-r) - 1.$$

As such,

$$\inf_s \mathcal{Q}_s = \inf_s -(\ell'_{\log} \circ \ell_{\log}^{-1})(n\widehat{\mathcal{R}}(u_s)) = \inf_s \left(1 - \exp(-n\widehat{\mathcal{R}}(u_s))\right)$$

$$= \left(1 - \exp(-n\inf_s \widehat{\mathcal{R}}(u_s))\right) > \left(1 - \exp(-0)\right) = 0,$$

meaning $\inf_s \mathcal{Q}_s > 0$ as desired.

2. If $\inf_s \widehat{\mathcal{R}}(u_s) = 0$, then we can choose a time $\tau$ with $\widehat{\mathcal{R}}(u_\tau) < \ell(0)$, and it follows that margins increase monotonically and $\widehat{\mathcal{R}}$ decreases monotonically (Lyu et al., 2021), and moreover by Lemma B.5 that $\|u_s\|$ is increasing and $\lim_s \|u_s\| = \infty$. Furthermore, for $\ell_{\exp}$, then $\sum_i q_i = 1$, whereas for $\ell_{\log}$, $\widehat{\mathcal{R}}(u_s) < \ell(0)$ (which holds for all $s \geq \tau$) implies $\sum_i q_i(s) \in [1, 2]$ (Ji & Telgarsky, 2019, Lemma 5.4). As such, for any $s \geq \tau$,

$$\frac{\mathrm{d}}{\mathrm{d}s} \ln \|u_s\|^2 = \frac{2\sum_i |\ell'_i| \langle u_s, \bar{\partial} p_i(u_s)\rangle}{\|u_s\|^2} = \frac{4\mathcal{Q}_s \sum_i q_i(s) p_i(u_s)}{\|u_s\|^2} \leq 8\mathcal{Q}_s,$$

whereby it follows that

$$\infty = \lim_t \ln \|u_t\|^2 = \int_\tau^\infty \|u_s\|^2 \, \mathrm{d}s \leq 8\int_\tau^\infty \mathcal{Q}_s \, \mathrm{d}s,$$

meaning $\int_\tau^\infty \mathcal{Q}_s \, \mathrm{d}s = \infty$, whereas $\int_0^\tau \mathcal{Q}_s \, \mathrm{d}s > 0$ via the analysis in the preceding case, and together $\int_0^\infty \mathcal{Q}_s \, \mathrm{d}s = \infty$.

Combining these two cases, then $\int_0^\infty \mathcal{Q}_s \, \mathrm{d}s = \infty$ unconditionally.

Now consider the second statement, with $\Phi, \tau, \gamma$ given. If $\liminf_t \gamma_t \geq \widehat{\gamma} > 0$, then $\lim_t \gamma_t$ is well-defined and positive via nondecreasing margins, and moreover $\|u\| \to \infty$ via Lemma B.5, and $0 \leq \limsup_t \widehat{\mathcal{R}}(u_t) \leq \limsup_t \ell(-\gamma_t \|w_t\|^L) = 0$. Alternatively, suppose contradictorily that $\liminf_t \gamma_t < \widehat{\gamma}$, and choose any $\epsilon \in (0, \widehat{\gamma}/4)$ so that $\liminf_t \gamma_t < \widehat{\gamma} - 3\epsilon$; noting that $\gamma_t$ is monotone once there exists some $\gamma_s > 0$, choose $t_1 \geq \tau$ large enough so that $\gamma_s \leq \widehat{\gamma} - 3\epsilon$ for all $s \geq t_1$. Next, note that $\|u\| \to \infty$ even in this situation (which may violate the conditions of Lemma B.5), since the assumptions $\Phi$ and the unconditional property $\int_0^\infty \mathcal{Q}_s \, \mathrm{d}s = \infty$ imply

$$\liminf_t \frac{1}{L} \ln \|u_t\| \geq \liminf_t \Phi(u_t) - \Phi(u_\tau) + \Phi(u_\tau) = \Phi(u_\tau) + \liminf_t \int_\tau^t \frac{\mathrm{d}}{\mathrm{d}s}\Phi(u_s) \, \mathrm{d}s$$

$$\geq \Phi(u_\tau) + \frac{1}{n}\liminf_t \int_\tau^t \widehat{\gamma}\mathcal{Q}_s \, \mathrm{d}s = \infty,$$

meaning $\|u_s\| \to \infty$; henceforth, choose $t_2 \geq t_1$ so that so that $\|u_s\|^2 \geq n/\epsilon$ for all $s \geq t_2$. It follows by Lemma B.6 and the assumption $L = 2$ that

$$0 \leq \liminf_t \left[\frac{1}{L} \ln \|u_t\| - \Phi(u_t)\right]$$

$$\leq \frac{1}{L} \ln \|u_{t_3}\| - \Phi(u_{t_3}) + \liminf_t \int_{t_3}^t \frac{\mathrm{d}}{\mathrm{d}s}\left[\frac{1}{4}\ln \|u_s\|^2 - \Phi(u_s)\right] \mathrm{d}s$$

$$= \frac{1}{L} \ln \|u_{t_3}\| - \Phi(u_{t_3}) + \liminf_t \int_{t_3}^t \frac{\mathrm{d}}{\mathrm{d}s}\left[\frac{\langle u_s, \dot{u}_s\rangle}{2\|u_s\|^2} - \Phi(u_s)\right] \mathrm{d}s$$

$$\leq \frac{1}{L} \ln \|u_{t_3}\| - \Phi(u_{t_3}) + \liminf_t \int_{t_3}^t \left[\frac{\mathcal{Q}_s(\gamma_s + n)}{n\|u_s\|^2} - \frac{1}{n}\mathcal{Q}_s\widehat{\gamma}\right] \mathrm{d}s$$

$$\leq \frac{1}{L} \ln \|u_{t_3}\| - \Phi(u_{t_3}) + \frac{1}{n}\liminf_t \int_{t_3}^t [-\epsilon\mathcal{Q}] \, \mathrm{d}s$$

$$= -\infty,$$

a contradiction, and since $\epsilon \in (0, \widehat{\gamma}/4)$ was arbitrary, it follows that $\liminf \gamma_t \geq \widehat{\gamma}$. $\qquad \square$

## C  PROOFS FOR SECTION 2

This section contains proofs for Section 2, all of which have a dependence on $\gamma_{\text{ntk}}$ rather than $\gamma_{\text{gl}}$. The SGD proofs will come first, as they are easier and serve as a warmup.

### C.1  SGD PROOFS

Before proceeding with the proof of Theorem 2.3, the following technical lemma (little more than an application of Freedman's inequality) will be a sufficient martingale concentration inequality for the test error bound.

**Lemma C.1** (Nearly identical to (Ji & Telgarsky, 2020b, Lemma 4.3)). *Define* $\mathcal{Q}(W) := \mathbb{E}_{x,y}|\ell'(p(x,y;W))|$ *and* $\mathcal{Q}_i(W) := |\ell'(p(x_i,y_i;W))|$. *Then* $\sum_{i<t}\left[\mathcal{Q}(W_i) - \mathcal{Q}_i(W_i)\right]$ *is a martingale difference sequence, and with probability at least* $1 - \delta$,

$$\sum_{i<t}\mathcal{Q}(W_i) \leq 4\sum_{i<t}\mathcal{Q}_i(W_i) + 4\ln(1/\delta),$$

*Proof.* This proof is essentially a copy of one due to Ji & Telgarsky (2020b, Lemma 4.3); that one is stated for the analog of $p_i$ used there, and thus needs to be re-checked.

Let $\mathcal{F}_i := \{((x_j, y_j)) : j < i\}$ denote the $\sigma$-field of all information until time $i$, whereby $x_i$ is independent of $\mathcal{F}_i$, whereas $w_i$ deterministic after conditioning on $\mathcal{F}_i$. Consequently, $\mathbb{E}\left[\mathcal{Q}(W_i) - \mathcal{Q}_i(W_i)|\mathcal{F}_i\right] = 0$, whereby $\sum_{i<t}\left[\mathcal{Q}(W_i) - \mathcal{Q}_i(W_i)\right]$ is a martingale difference sequence.

The high probability bound will now follow via a version of Freedman's inequality (Agarwal et al., 2014, Lemma 9). To apply this bound, the conditional variances must be controlled: noting that $|\ell'(z)| \in [0, 1]$, then $\mathcal{Q}(W_i) - \mathcal{Q}_i(W_i) \leq 1$, and since $\mathcal{Q}_i(W_i) \in [0, 1]$, then $\mathcal{Q}_i(W_i)^2 \leq \mathcal{Q}_i(W_i)$, and thus

$$\begin{aligned}
\mathbb{E}\left[\left(\mathcal{Q}(W_i) - \mathcal{Q}_i(W_i)\right)^2 \mid \mathcal{F}_i\right] &= \mathbb{E}\left[\mathcal{Q}_i(W_i)^2 \mid \mathcal{F}_i\right] - \mathcal{Q}(W_i)^2 \\
&\leq \mathbb{E}\left[\mathcal{Q}_i(W_i) \mid \mathcal{F}_i\right] - 0 \\
&= \mathcal{Q}(W_i).
\end{aligned}$$

As such, by the aforementioned version of Freedman's inequality (Agarwal et al., 2014, Lemma 9),

$$\begin{aligned}
\sum_{i<t}\left[\mathcal{Q}(W_i) - \mathcal{Q}_i(W_i)\right] &\leq (e-2)\sum_{i<t}\mathbb{E}\left[\left(\mathcal{Q}(W_i) - \mathcal{Q}_i(W_i)\right)^2 \mid \mathcal{F}_i\right] + \ln(1/\delta) \\
&\leq (e-2)\sum_{i<t}\mathcal{Q}(W_i) + \ln(1/\delta),
\end{aligned}$$

which rearranges to give

$$(3-e)\sum_{i<t}\mathcal{Q}(W_i) \leq \sum_{i<t}\mathcal{Q}_i(W_i) + \ln(1/\delta),$$

which gives the result after multiplying by 4 and noting $4(3 - e) \geq 1$. $\qquad\square$

With Lemma C.1 and the Gaussian concentration inequalities from Appendix B.3 in hand, the proof of Theorem 2.3 is as follows.

*Proof of Theorem 2.3.* Let $(w_j)_{j=1}^m$ be given with corresponding $(\bar{a}_j, \bar{v}_j) := \bar{\theta}_j := \theta(w_j)/\sqrt{m}$ (whereby $\|\bar{\theta}_j\| \leq 2$ by construction), and define

$$r := \frac{10\eta\sqrt{m}}{\gamma_{\text{ntk}}} \leq \frac{\gamma_{\text{ntk}}\sqrt{m}}{640}, \qquad R := 8r = \frac{80\eta\sqrt{m}}{\gamma_{\text{ntk}}} \leq \frac{\gamma_{\text{ntk}}\sqrt{m}}{80}, \qquad \overline{W} := r\bar{\theta} + W_0,$$

which implies $r \geq 1$, and $R \geq 1$, and $\eta \leq R/16$. For the remainder of the proof, rule out the $7\delta$ failure probability associated the second part of Lemma B.4, whereby simultaneously for every $\|W' - W_0\| \leq R$,

$$\min_i \left\langle \overline{W}, \hat{\partial} p_i(W') \right\rangle \geq \frac{r\gamma_{\mathrm{ntk}}\sqrt{m}}{2} - 160r^2 \geq \frac{r\gamma_{\mathrm{ntk}}\sqrt{m}}{4} = \frac{\gamma_{\mathrm{ntk}}^2 m}{2560} \geq \ln(t), \tag{C.1}$$

$$\min_i \left\langle \overline{\theta}, \hat{\partial} p_i(W') \right\rangle \geq \gamma_{\mathrm{ntk}}\sqrt{m} - \sqrt{32\ln(n/\delta)} - 4R - 4 \geq \gamma_{\mathrm{ntk}}\sqrt{m} - \frac{\gamma_{\mathrm{ntk}}\sqrt{m}}{8} - \frac{\gamma_{\mathrm{ntk}}\sqrt{m}}{10} \geq \frac{\gamma_{\mathrm{ntk}}\sqrt{m}}{2}, \tag{C.2}$$

and also $\|a_0\| \leq 2$ and $\|V_0\| \leq 2\sqrt{m}$.

The proof now proceeds as follows. Let $\tau$ denote the first iteration where $\|W_\tau - W_0\| \geq R$, whereby $\tau > 0$ and $\max_{s<\tau} \|W_s - W_0\| \leq R$. Assume contradictorily that $\tau \leq t$; it will be shown that this implies $\|W_\tau - W_0\| \leq R$.

Consider any iteration $s < \tau$. Expanding the square,

$$\begin{aligned}
\|W_{s+1} - \overline{W}\|^2 &= \|W_s - \eta\hat{\partial}\ell_s(W_s) - \overline{W}\|^2 \\
&= \|W_s - \overline{W}\|^2 - 2\eta \left\langle \hat{\partial}\ell_s(W_s), W_s - \overline{W} \right\rangle + \eta^2 \left\|\hat{\partial}\ell_s(W_s)\right\|^2 \\
&= \|W_s - \overline{W}\|^2 + 2\eta\ell_s'(W_s) \left\langle \hat{\partial}p_s(W_s), \overline{W} - W_s \right\rangle + \eta^2\ell_s'(W_s)^2 \left\|\hat{\partial}p_s(W_s)\right\|^2.
\end{aligned}$$

By convexity, $\|W_s - W_0\| \leq R$, and eq. (C.1),

$$\begin{aligned}
\ell_s'(W_s) \left\langle \hat{\partial}p_s(W_s), \overline{W} - W_s \right\rangle &= \ell_s'(W_s) \left( \left[ \left\langle \hat{\partial}p_s(W_s), \overline{W} \right\rangle - p_s(W_s) \right] - p_s(W_s) \right) \\
&\leq \ell_s \left( \left\langle \hat{\partial}p_s(W_s), \overline{W} \right\rangle - p_s(W_s) \right) - \ell_s(W_s) \\
&\leq \ln(1 + \exp(-\ln(t))) - \ell_s(W_s), \\
&\leq \frac{1}{t} - \ell_s(W_s),
\end{aligned}$$

which combined with the preceding display gives

$$\|W_{s+1} - \overline{W}\|^2 \leq \|W_s - \overline{W}\|^2 + 2\eta \left( \frac{1}{t} - \ell_s(W_s) \right) + \eta^2\ell_s'(W_s)^2 \left\|\hat{\partial}p_s(W_s)\right\|^2.$$

Since this inequality holds for any $s < \tau$, then applying the summation $\sum_{s<\tau}$ and rearranging gives

$$\|W_\tau - \overline{W}\|^2 + 2\eta \sum_{s<\tau} \ell_s(W_s) \leq \|W_0 - \overline{W}\|^2 + 2\eta + \sum_{s<\tau} \eta^2\ell_s'(W_s)^2 \left\|\hat{\partial}p_s(W_s)\right\|^2.$$

To simplify the last term, using $\|V_0\| \leq 2\sqrt{m}$ and $\|a_0\| \leq 2$ and $\|W_s - W_0\| \leq R$ gives

$$\begin{aligned}
\left\|\hat{\partial}p_s(W)\right\|^2 &= \|\sigma(V_s x_s)\|^2 + \left\|\sum_j e_j a_{i,j}\sigma'(v_{i,j}^\mathsf{T} x_s)x_s\right\|^2 \\
&\leq \left\|\sigma(V_s x_s)\right\|^2 + \|a_s\|^2 \\
&\leq 2\|V_s - V_0\|^2 + 2\|V_0\|^2 + 2\|a_s - a_0\|^2 + 2\|a_0\|^2 \\
&\leq 2R^2 + 8m + 8, \\
&\leq 10m,
\end{aligned}$$

and moreover the first term can be simplified via

$$\begin{aligned}
\|W_\tau - \overline{W}\|^2 &= \|W_\tau - W_0\|^2 - 2 \left\langle W_\tau - W_0, \overline{W} - W_0 \right\rangle + \|\overline{W} - W_0\|^2 \\
&\geq \|W_\tau - W_0\|^2 - 2r\|W_\tau - W_0\| + \|\overline{W} - W_0\|^2,
\end{aligned}$$

whereby combining these all gives

$$\|W_\tau - W_0\|^2 - 2r\|W_\tau - W_0\| + \|\overline{W} - W_0\|^2 + 2\eta \sum_{s<\tau} \ell_s(W_s)$$

$$\leq \|W_\tau - \overline{W}\|^2 + 2\eta \sum_{s<\tau} \ell_s(W_s)$$

$$\leq \|W_0 - \overline{W}\|^2 + 2\eta + \sum_{s<\tau} \eta^2 \ell'_s(W_s)^2 \left\|\hat{\partial} p_s(W_s)\right\|^2$$

$$\leq \|W_0 - \overline{W}\|^2 + 2\eta + 10\eta^2 m \sum_{s<\tau} |\ell'_s(W_s)|,$$

which after canceling and rearranging gives

$$\|W_\tau - W_0\|^2 + 2\eta \sum_{s<\tau} \ell_s(W_s) \leq 2r\|W_\tau - W_0\| + 2\eta + 10\eta^2 m \sum_{s<\tau} |\ell'_s(W_s)|.$$

To simplify the last term, note by eq. (C.2) that

$$\|W_\tau - W_0\| = \sup_{\|W\|\leq 1} \langle W, W_\tau - W_0 \rangle$$

$$\geq \frac{1}{2}\left\langle -\bar{\theta}, W_\tau - W_0 \right\rangle$$

$$= \frac{\eta}{2} \sum_{s<\tau} \left\langle -\bar{\theta}, \hat{\partial}\ell_s(W_s) \right\rangle$$

$$= \frac{\eta}{2} \sum_{s<\tau} |\ell'_s(W_s)| \left\langle \bar{\theta}, \hat{\partial} p_i(W_s) \right\rangle$$

$$\geq \frac{\eta}{2} \sum_{s<\tau} |\ell'_s(W_s)| \frac{\gamma_{\text{ntk}}\sqrt{m}}{2}, \tag{C.3}$$

and thus, by the choice of $R$, and since $\|W_\tau - W_0\| \geq 1$ and $\eta \leq R/16$,

$$\|W_\tau - W_0\|^2 + 2\eta \sum_{s<t} \ell_s(W_s) \leq 2r\|W_\tau - W_0\| + 2\eta + \frac{40\eta\sqrt{m}\|W_\tau - W_0\|}{\gamma_{\text{ntk}}}$$

$$\leq \left(\frac{R}{4} + \frac{R}{8} + \frac{R}{2}\right)\|W_\tau - W_0\|.$$

Dropping the term $2\eta \sum_{s<t} \ell_s(W_s) \geq 0$ and dividing both sides by $\|W_\tau - W_0\| \geq R > 0$ gives

$$\|W_\tau - W_0\| \leq \frac{R}{4} + \frac{R}{8} + \frac{R}{2} < R,$$

the desired contradiction, thus $\tau > t$ and all above derivations hold for all $s \leq t$.

To finish the proof, combining eq. (C.3) with $\|W_t - W_0\| \leq R = 80\eta\sqrt{m}/\gamma_{\text{ntk}}$ gives

$$\sum_{s<t} |\ell'_s(W_s)| \leq \frac{4\|W_t - W_0\|}{\eta\gamma_{\text{ntk}}\sqrt{m}} \leq \frac{320}{\gamma_{\text{ntk}}^2}.$$

Lastly, for the generalization bound, defining $\mathcal{Q}(W) := \mathbb{E}_{x,y}|\ell'(p(x,y;W))|$, discarding an additional $\delta$ failure probability, by Lemma C.1,

$$\sum_{s<t} \mathcal{Q}(W_s) \leq 4\ln(1/\delta) + 4\sum_{s<t} |\ell'_s(W_s)| \leq 4\ln(1/\delta) + \frac{1280}{\gamma_{\text{ntk}}^2}.$$

Since $\mathbb{1}[p_s(W_s) \leq 0] \leq 2|\ell'_s(W_s)|$, the result follows.

It remains to argue that $\|W_1 - W_0\|$ is large; to this end, it already holds by instantiating eq. (C.3) with $\tau = 1$ that

$$\|W_1 - W_0\| \geq \frac{\eta\gamma_{\text{ntk}}|\ell'_0(W_0)|\sqrt{m}}{4},$$

so it only remains to show that $|\ell_0'(W_0)|$ is not too small. By Lemma B.3, discarding an additional failure probability $\delta$, it holds that $|F(x_0; W_0)| \leq 4\ln(1/\delta)$, and therefore

$$|\ell_0'(W_0)| = \frac{1}{1 + \exp(p_0(W_0))} \geq \frac{1}{1 + 1/\delta^4} \geq \delta^4,$$

which combines to give $\|W_1 - W_0\| \geq \frac{\eta\gamma_{\text{ntk}}\delta^4\sqrt{m}}{8}$ as desired. $\qquad\square$

## C.2  GF PROOFS

This section culminates in the proof of Theorem 2.1, which is broken into a few main lemmas: Lemma C.3 first controls the empirical risk $\widehat{\mathcal{R}}$ similarly to the proof of Theorem 2.3, then Lemma C.4 establishes large margins, whereby Lemma C.6 develops a suitable Rademacher complexity bound, which combine to quickly give the proof of Theorem 2.1.

Before proceeding with the main proofs, the following technical lemma is used to convert a bound on $\ell'$ to a bound on $\ell$.

**Lemma C.2.** *For $\ell \in \{\ell_{\log}, \ell_{\exp}\}$, then $|\ell'(z)| \leq 1/8$ implies $\ell(z) \leq 2|\ell'(z)|$.*

*Proof.* If $\ell = \ell_{\exp}$, then $\ell' = -\ell$, and thus $\ell(z) \leq 2|\ell'(z)|$ automatically. If $\ell(z) = \ell_{\log}$, the logistic loss, then $|\ell'(z)| \leq 1/8$ implies $z \geq 2$. By the concavity of $\ln(\cdot)$, for any $z \geq 2$, since $1 + e^{-z} \leq 7/6$, then

$$\ell(z) = \ln(1 + e^{-z}) \leq e^{-z} \leq \frac{(7/6)e^{-z}}{1 + e^{-z}} \leq 2|\ell'(z)|,$$

thus completing the proof. $\qquad\square$

Next comes the proof of Lemma C.3, which follows the same proof plan as Theorem 2.3.

**Lemma C.3.** *Suppose the data distribution satisfies Assumption 1.2 for some $\gamma_{\text{ntk}} > 0$, let time $t$ be given, and suppose width $m$ satisfies*

$$m \geq \left(\frac{640\ln(t/\delta)}{\gamma_{\text{ntk}}}\right)^2.$$

*Then, with probability at least $1 - 7\delta$, the GF curve $(W_s)_{s \in [0,t]}$ on empirical risk $\widehat{\mathcal{R}}$ with loss $\ell \in \{\ell_{\log}, \ell_{\exp}\}$ satisfies*

$$\widehat{\mathcal{R}}(W_t) \leq \frac{1}{5t}, \qquad\qquad \text{(training error bound)},$$

$$\sup_{s<t} \|W_s - W_0\| \leq \frac{\gamma_{\text{ntk}}\sqrt{m}}{80}, \qquad\qquad \text{(norm bound)}.$$

Note that this bound is morally equivalent to the SGD bound in Theorem 2.3 after accounting for the $\gamma_{\text{ntk}}^2$ "units" arising from the step size.

*Proof of Lemma C.3.* This proof is basically identical to the SGD in Theorem 2.3. Despite this, proceeding with amnesia, let rows $(w_j)_{j=1}^m$ of $W_0$ be given with corresponding $(\bar{a}_j, \bar{v}_j) := \bar{\theta}_j := \theta(w_j)/\sqrt{m}$ (whereby $\|\bar{\theta}_j\| \leq 2$ by construction), and define

$$r := \frac{\gamma_{\text{ntk}}\sqrt{m}}{640}, \qquad R := 8r = \frac{\gamma_{\text{ntk}}\sqrt{m}}{80}, \qquad \overline{W} := r\bar{\theta} + W_0,$$

with immediate consequences that $r \geq 1$ and $R \geq 8$. For the remainder of the proof, rule out the $7\delta$ failure probability associated with the second part of Lemma B.4, whereby simultaneously for every $\|W' - W_0\| \leq R$,

$$\min_i \left\langle \overline{W}, \hat{\partial}p_i(W') \right\rangle \geq \frac{r\gamma_{\text{ntk}}\sqrt{m}}{2} - 160r^2 \geq \frac{r\gamma_{\text{ntk}}\sqrt{m}}{4} = \frac{\gamma_{\text{ntk}}^2 m}{2560} \geq \ln(t), \qquad (\text{C.4})$$

$$\min_i \left\langle \bar{\theta}, \hat{\partial}p_i(W') \right\rangle \geq \gamma_{\text{ntk}}\sqrt{m} - \sqrt{32\ln(n/\delta)} - 4R - 4 \geq \gamma_{\text{ntk}}\sqrt{m} - \frac{\gamma_{\text{ntk}}\sqrt{m}}{8} - \frac{\gamma_{\text{ntk}}\sqrt{m}}{10} \geq \frac{\gamma_{\text{ntk}}\sqrt{m}}{2}.$$
$$(\text{C.5})$$

The proof now proceeds as follows. Let $\tau$ denote the earliest time such that $\|W_\tau - W_0\| = R$; since $W_s$ traces out a continuous curve and since $R > 0 = \|W_0 - W_0\|$, this quantity is well-defined. As a consequence of the definition, $\sup_{s < \tau} \|W_s - W_0\| \leq R$. Assume contradictorily that $\tau \leq t$; it will be shown that this implies $\|W_\tau - W_0\| < R$.

By the fundamental theorem of calculus (and the chain rule for Clarke differentials), convexity of $\ell$, and since $\|W_s - W_0\| \leq R$ holds for $s \in [0, \tau)$, which implies eq. (C.4) holds,

$$
\begin{aligned}
\|W_\tau - \overline{W}\|^2 - \|W_0 - \overline{W}\|^2 &= \int_0^\tau \frac{\mathrm{d}}{\mathrm{d}s} \|W_s - \overline{W}\|^2 \, \mathrm{d}s \\
&= \int_0^\tau 2 \left\langle \dot{W}_s, W_s - \overline{W} \right\rangle \mathrm{d}s \\
&= \frac{2}{n} \int_0^\tau \sum_i \ell_i'(W_s) \left\langle \bar{\partial} p_i(W_s), W_s - \overline{W} \right\rangle \mathrm{d}s \\
&= \frac{2}{n} \int_0^\tau \sum_i \ell_i'(W_s) \left( \left[ \left\langle \hat{\partial} p_i(W_s), \overline{W} \right\rangle - p_i(W_s) \right] - p_i(W_s) \right) \mathrm{d}s \\
&\leq \frac{2}{n} \int_0^\tau \sum_i \left( \ell_i \left( \left\langle \hat{\partial} p_i(W_s), \overline{W} \right\rangle - p_i(W_s) \right) - \ell_i(W_s) \right) \mathrm{d}s \\
&\leq \frac{2}{n} \int_0^\tau \sum_i \left( \frac{1}{t} - \ell_i(W_s) \right) \mathrm{d}s \\
&\leq 2 - 2 \int_0^\tau \widehat{\mathcal{R}}(W_s) \, \mathrm{d}s.
\end{aligned}
$$

To simplify the left hand side,

$$
\|W_\tau - \overline{W}\|^2 - \|W_0 - \overline{W}\|^2 = \|W_\tau - W_0\|^2 - 2 \left\langle W_\tau - W_0, \overline{W} - W_0 \right\rangle \geq \|W_\tau - W_0\|^2 - 2r \|W_\tau - W_0\|,
$$

which after combining, rearranging, and using $r \geq 1$ and $\|W_\tau - W_0\| \geq R \geq 1$ gives

$$
\|W_\tau - W_0\|^2 + 2 \int_0^\tau \mathcal{R}(W_s) \, \mathrm{d}s \leq 2 + 2r \|W_\tau - W_0\| \leq 4r \|W_\tau - W_0\|,
$$

which implies

$$
\|W_\tau - W_0\| \leq 2r = \frac{R}{2} < R,
$$

a contradiction since $W_\tau$ is well-defined as the earliest time with $\|W_\tau - W_0\| = R$, which thus contradicts $\tau \leq t$. As such, $\tau \geq t$, and all of the preceding inequalities follows with $\tau$ replaced by $t$.

To obtain an error bound, similarly to the key perceptron argument before, using eq. (C.5),

$$
\begin{aligned}
\|W_t - W_0\| &= \sup_{\|W\| \leq 1} \langle W, W_t - W_0 \rangle \\
&\geq \frac{1}{2} \left\langle -\bar{\theta}, W_t - W_0 \right\rangle \\
&= \frac{1}{2} \left\langle -\bar{\theta}, \int_0^t \dot{a}_s \, \mathrm{d}s \right\rangle \\
&= \frac{1}{2n} \int_0^t \sum_i |\ell_i'(W_s)| \left\langle \bar{\theta}, \hat{\partial} p_i(W_s) \right\rangle \mathrm{d}s \\
&\geq \frac{\gamma_{\text{ntk}} \sqrt{m}}{4n} \int_0^t \sum_i |\ell_i'(W_s)| \, \mathrm{d}s,
\end{aligned}
$$

which implies

$$
\frac{1}{n} \int_0^t \sum_i |\ell_i'(W_s)| \, \mathrm{d}s \leq \frac{4 \|W_t - W_0\|}{\gamma_{\text{ntk}} \sqrt{m}} \leq \frac{1}{20},
$$

and in particular

$$\inf_{s\in[0,t]} \frac{1}{n}\sum_i |\ell'_i(W_s)| \le \frac{1}{tn}\int_0^t \sum_i |\ell'_i(W_s)|\,\mathrm{d}s \le \frac{4\|W_t - W_0\|}{t\gamma_{\mathrm{ntk}}\sqrt{m}} \le \frac{1}{20t}$$

and so there exists $k \in [0, t]$ with

$$\frac{1}{n}\sum_i |\ell'_i(W_k)| \le \frac{1}{10t}.$$

Since this also implies $\max_i |\ell'_i(W_k)| \le n/(10t) \le 1/10$, it follows by Lemma C.2 that $\widehat{\mathcal{R}}(W_k) \le 1/(5t)$, and the claim also holds for $t' \ge t$ since the empirical risk is nonincreasing with gradient flow. □

Next is the explicit maximum margin guarantee, which was missing from the SGD analysis.

**Lemma C.4.** *Suppose the data distribution satisfies Assumption 1.2 with margin $\gamma_{\mathrm{ntk}} > 0$ and parameter mapping $\theta$, and let $((x_i, y_i))_{i=1}^n$ be an iid draw. Let $(W_s)_{s\ge 0}$ denote the GF curve resulting from loss $\ell \in \{\ell_{\log}, \ell_{\exp}\}$. Suppose the width $m$ satisfies*

$$m \ge \frac{256\ln(n/\delta)}{\gamma_{\mathrm{ntk}}^2},$$

*fix a distance parameter $R := \gamma_{\mathrm{ntk}}\sqrt{m}/32$, and let time $\tau$ be given so that $\|W_\tau - W_0\| \le R/2$ and $\widehat{\mathcal{R}}(W_\tau) < \ell(0)/n$. Then, with probability at least $1 - 7\delta$, there exists a time $t$ with $\|W_t - W_0\| = R$ so that for all $s \ge t$,*

$$\|W_s - W_0\| \ge R \qquad \text{and} \qquad \mathring{\gamma}(W_s) \ge \frac{\gamma_{\mathrm{ntk}}^2}{4096},$$

*and moreover the rebalanced iterate $\widehat{W}_t := (a_t/\sqrt{\gamma_{\mathrm{ntk}}}, V_t\sqrt{\gamma_{\mathrm{ntk}}})$ satisfies $p(x, y; W_t) = p(x, y; \widehat{W}_t)$ for all $(x, y)$, and*

$$\frac{\widetilde{\gamma}(W_t)}{2\|a_t\|\cdot\|V_t\|} \ge \mathring{\gamma}(\widehat{W}_t) \ge \frac{\gamma_{\mathrm{ntk}}}{4096}.$$

Before discussing the proof, a few remarks are in order. Firstly, the final large margin iterate $W_t$ is stated as explicitly achieving some distance from initialization; needing such a claim is unsurprising, as the margin definition requires a lot of motion in a good direction to clear the noise in $W_0$. In particular, it is unsurprising that moving $\mathcal{O}(\sqrt{m})$ is needed to achieve a good margin, given that the initial weight norm is $\mathcal{O}(\sqrt{m})$; analogously, it is not surprising that Lemma C.3 can not be used to produce a meaningful lower bound on $\mathring{\gamma}(W_\tau)$ directly. Lastly, while these comments seem natural when normalizing by $\|W_t\|^2$, the normalization $\|a_t\|\cdot\|V_t\|$ does not obviously have these deficiencies.

Another thing to highlight is that the use of Lemma C.3 for warm start is only needed for $\ell_{\log}$, and not for $\ell_{\exp}$; it is unclear how this discrepancy translates to practice, where $\ell_{\log}$ dominates.

*Proof of Lemma C.4.* By the second part of Lemma B.4, with probability at least $1 - 7\delta$, simultaneously $\|a\| \le 2$, and $\|V\| \le 2\sqrt{m}$, and for any $\|W' - W_0\| \le R$, then

$$\min_i \left\langle \overline{\theta}, \hat{\partial}p_i(W') \right\rangle \ge \gamma_{\mathrm{ntk}}\sqrt{m} - \left[\sqrt{32\ln(n/\delta)} + 8R + 4\right] \ge \frac{\gamma_{\mathrm{ntk}}\sqrt{m}}{2},$$

where $\overline{\theta}_j := \theta(w_j)/\sqrt{m}$ as usual, and $\|\overline{\theta}\| \le 2$; for the remainder of the proof, suppose these bounds, and discard the corresponding $7\delta$ failure probability. Moreover, for any $W'$ with $\widehat{\mathcal{R}}(W') < \ell(0)/n$ and $\|W' - W_0\| \le R$, as a consequence of the preceding lower bound and also the property $\sum_i q_i(W') \ge 1$ Ji & Telgarsky (2019, Lemma 5.4, first part, which does not depend on linear

predictors),

$$\left\|\bar{\partial}\widetilde{\gamma}(W')\right\| = \sup_{\|W\|\leq 1}\left\langle W, \bar{\partial}\widetilde{\gamma}(W')\right\rangle$$

$$\geq \frac{1}{2}\left\langle \bar{\theta}, \sum_i q_i \bar{\partial}p_i(W')\right\rangle$$

$$= \frac{1}{2}\sum_i q_i\left\langle \bar{\theta}, \bar{\partial}p_i(W')\right\rangle$$

$$\geq \frac{\gamma_{\mathrm{ntk}}\sqrt{m}}{4}\sum_i q_i$$

$$\geq \frac{\gamma_{\mathrm{ntk}}\sqrt{m}}{4}.$$

Now consider the given $W_\tau$ with $\widehat{\mathcal{R}}(W_\tau) < \ell(0)/n$ and $\|W_\tau - W_0\| \leq R/2$. Since $s \mapsto W_s$ traces out a continuous curve and since norms grow monotonically and unboundedly after time $\tau$ (cf. Lemma B.5), then there exists a unique time $r$ with $\|W_t - W_0\| = R$. Furthermore, since $\widehat{\mathcal{R}}$ is nonincreasing throughout gradient flow, then $\widehat{\mathcal{R}}(W_s) < \ell(0)/n$ holds for all $s \in [\tau, t]$. Then

$$\widetilde{\gamma}(W_t) - \widetilde{\gamma}(W_\tau) = \int_\tau^t \left\langle \bar{\partial}\widetilde{\gamma}(W_s), \dot{W}_s\right\rangle \mathrm{d}s$$

$$= \int_\tau^t \|\bar{\partial}\widetilde{\gamma}(W_s)\| \cdot \|\dot{W}_s\| \, \mathrm{d}s$$

$$\geq \frac{\gamma_{\mathrm{ntk}}\sqrt{m}}{4}\int_\tau^t \|\dot{W}_s\| \, \mathrm{d}s$$

$$\geq \frac{\gamma_{\mathrm{ntk}}\sqrt{m}}{4}\left\|\int_\tau^t \dot{W}_s\right\| \mathrm{d}s$$

$$= \frac{\gamma_{\mathrm{ntk}}\sqrt{m}}{4}\|W_t - W_\tau\| \, \mathrm{d}s$$

$$\geq \frac{\gamma_{\mathrm{ntk}}R\sqrt{m}}{8}$$

$$\geq \frac{\gamma_{\mathrm{ntk}}^2 m}{256}.$$

Since $\|W_0\| \leq 3\sqrt{m}$, thus $\|W_t\| \leq 3\sqrt{m} + \gamma_{\mathrm{ntk}}\sqrt{m}/32 \leq 4\sqrt{m}$, and the normalized margin satisfies

$$\mathring{\gamma}(W_t) \geq \frac{\widetilde{\gamma}(W_\tau)}{\|W_t\|^2} + \frac{1}{\|W_t\|^2}\int_\tau^t \frac{\mathrm{d}}{\mathrm{d}s}\widetilde{\gamma}(W_s)\,\mathrm{d}s \geq 0 + \frac{\gamma_{\mathrm{ntk}}^2 m/256}{16m} = \frac{\gamma_{\mathrm{ntk}}^2}{4096}.$$

Furthermore, it holds that $\mathring{\gamma}(W_s) \geq \mathring{\gamma}(W_t)$ for all $s \geq t$ (Lyu & Li, 2019), which completes the proof for $W_t$ under the standard parameterization.

Now consider the rebalanced parameters $\widehat{W}_t := (a_t/\sqrt{\gamma_{\mathrm{ntk}}}, V_t\sqrt{\gamma_{\mathrm{ntk}}})$; since $m \geq 256/\gamma_{\mathrm{ntk}}^2$, which means $16 \leq \gamma_{\mathrm{ntk}}\sqrt{m}$, then

$$\|a_t\| \leq \|a_0\| + \|a_t - a_0\| \leq 2 + R \leq \frac{\gamma_{\mathrm{ntk}}\sqrt{m}}{8} + \frac{\gamma_{\mathrm{ntk}}\sqrt{m}}{32} \leq \frac{\gamma_{\mathrm{ntk}}\sqrt{m}}{4},$$

$$\|V_t\| \leq \|V_0\| + \|V_t - V_0\| \leq 2\sqrt{m} + R \leq 3\sqrt{m},$$

then the rebalanced parameters satisfy

$$\|\widehat{W}_t\| \leq \|a_t/\sqrt{\gamma_{\mathrm{ntk}}}\| + \|V_t\sqrt{\gamma_{\mathrm{ntk}}}\| \leq \frac{\sqrt{\gamma_{\mathrm{ntk}}m}}{4} + 3\sqrt{\gamma_{\mathrm{ntk}}m} \leq 4\sqrt{\gamma_{\mathrm{ntk}}m},$$

and thus, for any $(x, y)$, since

$$p(x, y; W_t) = \sum_j a_j(t)\sigma(v_j(t)^\intercal x) = \sum_j \frac{a_j(t)}{\sqrt{\gamma_{\mathrm{ntk}}}}\sigma(\sqrt{\gamma_{\mathrm{ntk}}}v_j(t)^\intercal x) = p(x, y; \widehat{W}_t),$$

then

$$\mathring{\gamma}(\widehat{W}_t) = \min_i \frac{p_i(\widehat{W}_t)}{\|\widehat{W}_t\|^2} = \min_i \frac{p_i(W_t)}{\|\widehat{W}_t\|^2} \geq \frac{\widetilde{\gamma}^2 m/256}{16\gamma_{\text{ntk}} m} \geq \frac{\gamma_{\text{ntk}}}{4096},$$

and lastly to complete the proof note by AM-GM that

$$\|\widehat{W}_t\|^2 = \frac{1}{\gamma_{\text{ntk}}}\|a_t\|^2 + \gamma_{\text{ntk}}\|V_t\|^2 \geq 2\|a_t\| \cdot \|V_t\|,$$

whereby $\mathring{\gamma}(\widehat{W}_t) = \frac{\widetilde{\gamma}(W_t)}{\|\widehat{W}_t\|^2} \leq \frac{\widetilde{\gamma}(W_t)}{2\|a_t\| \cdot \|V_t\|}$. $\qquad\square$

Next comes a margin-based Rademacher complexity bound; as Rademacher complexity has not been used or defined within this work, here is a brief description of the main definition, with further detail deferred tp standard references (Shalev-Shwartz & Ben-David, 2014). First, for a given set of vectors $V \subseteq \mathbb{R}^n$, the *Rademacher complexity* $\text{Rad}(V)$ is

$$\text{Rad}(V) = \frac{1}{n}\mathbb{E}_\epsilon \sup_{u \in V} \langle \epsilon, u \rangle,$$

where $\epsilon \in \{\pm 1\}^n$ has iid Rademacher coordinates, meaning $\Pr[\epsilon_i = +1] = \frac{1}{2} = \Pr[\epsilon_i = -1]$. The set $V$ will typically be the set of outputs of some class of predictors $\mathcal{G}$ on a finite sample $\mathcal{X} = (x_i)_{i=1}^n$ of size $n$, using the notation

$$\mathcal{G}_{|\mathcal{X}} = \left\{ \big(g(x_1), \ldots, g(x_n)\big) : g \in \mathcal{G} \right\} \subseteq \mathbb{R}^n.$$

Our bound below will replace $\mathcal{G}$ with a variety of bounded two-layer networks of any width. This bound can be viewed as a strengthening of the proofs of (Vardi et al., 2022), where the bound here holds for all widths simultaneously (with no dependence on width), and is normalized by the tighter quantity $\sum_j \|a_j v_j\|$.

**Lemma C.5.** *For any $B \geq 0$ and any $\mathcal{X} = (x_i)_{i=1}^n$ with $\|x_i\| \leq 1$,*

$$\text{Rad}\left(\left\{x \mapsto F(x; W) : m \geq 0, W \in \mathbb{R}^{m \times (d+1)}, \|W\|^2 \leq 2B\right\}_{|\mathcal{X}}\right)$$

$$\leq \text{Rad}\left(\left\{x \mapsto F(x; W) : m \geq 0, (a, V) = W \in \mathbb{R}^{m \times (d+1)}, \|a\| \cdot \|V\| \leq B\right\}_{|\mathcal{X}}\right)$$

$$\leq \text{Rad}\left(\left\{x \mapsto F(x; W) : m \geq 0, (a, V) = W \in \mathbb{R}^{m \times (d+1)}, \sum_j \|a_j v_j\| \leq B\right\}_{|\mathcal{X}}\right)$$

$$\leq \frac{2B}{\sqrt{n}}.$$

*Proof.* The first two inequalities are easier, and follow by set inclusion. In detail, note for any fixed $W = (a, V)$, by Cauchy-Schwarz and AM-GM, that

$$\sum_j \|a_j v_j\| = \sum_j |a_j| \cdot \|v_j\| \leq \|a\| \cdot \|V\| \leq \frac{1}{2}\left(\|a\|^2 + \|V\|^2\right) = \frac{\|W\|^2}{2},$$

which implies for each $m$ the inclusions

$$\left\{W \in \mathbb{R}^{m \times (d+1)} : \|W\|^2 \leq 2B\right\} \subseteq \left\{(a, V) = W \in \mathbb{R}^{m \times (d+1)} : \|a\| \cdot \|V\| \leq B\right\}$$

$$\subseteq \left\{(a, V) = W \in \mathbb{R}^{m \times (d+1)} : \sum_j \|a_j v_j\| \leq B\right\},$$

which in turn implies the first two Rademacher inequalities in the statement since Rademacher complexity can not decrease with the growth of sets.

For the final inequality, recall the definition of *symmetric convex hull* sconv($\cdot$) as used throughout Rademacher complexity (Shalev-Shwartz & Ben-David, 2014):

$$\mathrm{sconv}(S) := \left\{ \sum_{j=1}^{m} p_j u_j \; : \; m \geq 0, p \in \mathbb{R}^m, \|p\|_1 \leq 1, u_j \in S \right\}.$$

Then, recalling the notation $\widetilde{a}_j$ and $\widetilde{v}_j$ for normalized counterparts to $a_j$ and $v_j$, the set of vectors $U$ in the final Rademacher term can be written

$$
\begin{aligned}
U &:= \left\{ x \mapsto \sum_j a_j \sigma(v_j^\mathsf{T} x) \; : \; m \geq 0, W \in \mathbb{R}^{m \times (d+1)}, \sum_j \|a_j v_j\| \leq B \right\}_{|\mathcal{X}} \\
&= \left\{ x \mapsto \sum_j \|a_j v_j\| \widetilde{a} \sigma(\widetilde{v}_j^\mathsf{T} x) \; : \; m \geq 0, W \in \mathbb{R}^{m \times (d+1)} \right\}_{|\mathcal{X}} \\
&= \left\{ x \mapsto \left( \sum_k \|a_k v_k\| \right) \sum_j \frac{\|a_j v_j\|}{\sum_k \|a_k v_k\|} \widetilde{a} \sigma(\widetilde{v}_j^\mathsf{T} x) \; : \; m \geq 0, W \in \mathbb{R}^{m \times (d+1)}, \sum_j \|a_j v_j\| \leq B \right\}_{|\mathcal{X}} \\
&= B \left\{ x \mapsto \sum_j p_j \sigma(\widetilde{v}_j^\mathsf{T} x) \; : \; m \geq 0, W \in \mathbb{R}^{m \times (d+1)}, p \in \mathbb{R}^m, \|p\|_1 \leq 1 \right\}_{|\mathcal{X}} \\
&= B \cdot \mathrm{sconv}\left( \left\{ x \mapsto \sigma(v^\mathsf{T} x) : \|v\|_2 = 1 \right\}_{|\mathcal{X}} \right),
\end{aligned}
$$

and by standard rules of Rademacher complexity (Shalev-Shwartz & Ben-David, 2014),

$$
\begin{aligned}
\mathrm{Rad}(U) &\leq B \cdot \mathrm{Rad}\left( \mathrm{sconv}\left( \left\{ x \mapsto \sigma(v^\mathsf{T} x) : \|v\|_2 = 1 \right\}_{|\mathcal{X}} \right) \right) \\
&\leq 2B \cdot \mathrm{Rad}\left( \left\{ x \mapsto \sigma(v^\mathsf{T} x) : \|v\|_2 = 1 \right\}_{|\mathcal{X}} \right) \\
&\leq \frac{2B}{\sqrt{n}}.
\end{aligned}
$$

$\square$

The large margin generalization bound is now an immediate consequence of Lemma C.5 and a refined margin-based Rademacher bound due to Srebro et al. (2010, Theorem 5). This bound will use the refined normalized margin

$$\gamma_1(W) := \frac{\min_i p_i(W)}{\sum_j \|a_j v_j\|}, \tag{C.6}$$

where Cauchy-Schwarz and AM-GM imply $\gamma_1(W) \geq \frac{\min_i p_i(W)}{\|a\| \cdot \|V\|} \geq 2\gamma(W)$ as in the proof of Lemma C.5.

**Lemma C.6.** *With probability at least $1 - \delta$ over the draw of $((x_i, y_i))_{i=1}^{n}$, for every width $m$, every choice of weights $(a, V) = W \in \mathbb{R}^{m \times (d+1)}$ with $\gamma_1(W) > 0$ (cf. eq. (C.6)) satisfies*

$$\Pr[p(x, y; W) \leq 0] \leq \mathcal{O}\left( \frac{\ln(n)^3}{n \gamma_1(W)^2} + \frac{\ln \frac{1}{\delta}}{n} \right).$$

*Proof.* Combining a refined Rademacher-based margin bound due to (Srebro et al., 2010, Theorem 5) with the 2-layer Rademacher complexity estimate from Lemma C.5 gives, with probability at least $1 - \delta$, for every margin level $\gamma_2 > 0$, for every $W \in \mathbb{R}^{m \times (d+1)}$ with $\gamma_1(W) \geq \gamma_2$, defining

$$U_{\gamma_2} := \{x \mapsto F(x; W)/\sum_j \|a_j v_j\| : \gamma_1(W) \ge \gamma_2\}_{|\mathcal{X}},$$

$$\Pr[p(x, y; W) \le 0] = \mathcal{O}\left(\frac{\ln(n)^3}{\gamma_2^2}\mathrm{Rad}(U_{\gamma_2})^2 + \frac{\ln\ln\frac{1}{\gamma_2} + \ln\frac{1}{\delta}}{n}\right)$$

$$= \mathcal{O}\left(\frac{\ln(n)^3}{\gamma_2^2}\left(\frac{1}{n}\right) + \frac{\ln\frac{1}{\delta}}{n}\right),$$

which implies the desired statement. $\square$

Thanks to Lemmas C.3 and C.4, the proof of Theorem 2.1 is now immediate.

*Proof of Theorem 2.1.* As in the statement, define $R := \gamma_{\mathrm{ntk}}\sqrt{m}/32$; the analysis now uses two stages. The first stage is handled by Lemma C.3 run until time $\tau := n$, whereby, with probability at least $1 - 7\delta$,

$$\widehat{\mathcal{R}}(W_\tau) \le \frac{1}{5n} < \frac{\ell(0)}{n}, \qquad \|W_\tau - W_0\| \le \frac{\gamma_{\mathrm{ntk}}\sqrt{m}}{80} \le \frac{R}{2}.$$

The second stage now follows from Lemma C.4: since $W_\tau$ as above satisfies all the conditions of Lemma C.4, there exists $W_t$ with $\|W_t - W_0\| = R$, and $\widetilde{\gamma}(W_t)/(\|a_t\| \cdot \|V_t\|) \ge \gamma_{\mathrm{ntk}}/2048$, and since even these mixed-norm margins are nondecreasing (Lyu & Li, 2019, Section H), the claim also holds for all $W_s$ with $s \ge t$, and the generalization bound follows from Lemma C.6, using $\sum_j \|a_j v_j\| \le \|a\| \cdot \|V\|$ and $\min_i p_i(W) \ge \widetilde{\gamma}(W)$, which holds for both $\{\ell_{\log}, \ell_{\exp}\}$. Lastly, since $\|W_s\| \to \infty$ and since $\lim_{s \to \infty} \|a_s\|/\|V_s\| = 1$ (Du et al., 2018a), it follows that $\lim_{s \to \infty} \|W_s\|^2/(\|a_s\| \cdot \|V_s\|) = 2$, which gives the final claim. $\square$

Lastly, the proof of Corollary 2.2, giving a simple construction where GF escapes bad KKT points.

*Proof of Corollary 2.2.* First it is shown that the provided choice of $(a, V)$ with $a_j = 1$ and $v_j = (1, 0)$ is a KKT direction. With probability $1 - 2^{1-n}$, both elements of the support of the distribution are sampled, and for convenience reorder the sampled data so that $x_1 = z_1$, and $x_2 = z_2$, and the other data are arbitrary (though $y_i = +1$ for all examples). It will be shown that the choice $\lambda_1 = \lambda_2 = 1/(2\gamma_0)$ and $\lambda_i = 0$ for $i \ge 3$ are a valid choice of Lagrange multipliers, certifying that $(a, V)$ is a KKT direction. Firstly, derivative condition is easy to check since the Clarke differential is evaluated where the ReLU is differentiable, and it holds directly that

$$\begin{aligned}
a_j &= 1 \\
&= \frac{\gamma_0}{2\gamma_0} + \frac{\gamma_0}{2\gamma_0} \\
&= \lambda_1 \sigma(v_j^\mathsf{T} x_1) + \lambda_2 \sigma(v_j^\mathsf{T} x_2), \\
v_j &= (1, 0) \\
&= \left(\frac{\gamma_0}{2\gamma_0} + \frac{\gamma_0}{2\gamma_0}, \frac{\sqrt{1 - \gamma_0^2}}{2\gamma_0} - \frac{\sqrt{1 - \gamma_0^2}}{2\gamma_0}\right) \\
&= \lambda_1 a_j x_1 + \lambda_2 a_j x_2 \\
&= \lambda_1 a_j \nabla_v \sigma(v_j^\mathsf{T} x_1) + \lambda_2 a_j \nabla_v \sigma(v_j^\mathsf{T} x_2).
\end{aligned}$$

Lastly, note that $p_1(W) = m\gamma_0 = p_2(W)$, therefore the rescaling $\widehat{W} := W/\sqrt{m\gamma_0}$ correctly satisfies $p_1(\widehat{W}) = 1 = p_2(\widehat{W})$, whereby $\widehat{W}$ is a KKT point with margin $m\gamma_0/\|W\|^2 = \gamma_0/2$, and $W$ is a KKT direction with margin $\gamma_0/2$.

For the GF guarantee, it suffices to provide a quick estimate for $\gamma_{\mathrm{ntk}}$ and invoke Lemma C.3. Specifically, consider the rather loose but convenient weight mapping

$$\theta(a, v) := \begin{cases} 0 & \|(a, v)\| \ge 2, \\ (0, \mathrm{sgn}(a)(0, 1)) & v_2 \ge |v_1|, \\ (0, \mathrm{sgn}(a)(0, -1)) & v_2 < -|v_1|. \end{cases}$$

Then, for the point $z_1$, since $\sqrt{1 - \gamma_0^2} > \gamma_0$, it follows that

$$\mathbb{E}\left\langle \theta((a, v)), \bar{\partial} p(z_1, +1; (a, v)) \right\rangle$$

$$= \mathbb{E}\mathbb{1}[v_2 \geq |v_1| \wedge \|(a, v)\| \leq 2]\left(0 + |a|\sigma'(\gamma_0 v_1 + \sqrt{1 - \gamma_0^2}v_2)\sqrt{1 - \gamma_0^2}\right)$$

$$+ \mathbb{E}\mathbb{1}[v_2 < 0 \wedge \|(a, v)\| \leq 2]\left(0 - |a|\sigma'(\gamma_0 v_1 + \sqrt{1 - \gamma_0^2}v_2)\sqrt{1 - \gamma_0^2}\right)$$

$$\geq \mathbb{E}|a|\mathbb{1}[v_2 \geq |v_1| \wedge \|(a, v)\| \leq 2]\sqrt{1 - \gamma_0^2}$$

$$\geq \frac{\sqrt{1 - \gamma_0^2}}{16},$$

where the last step follows from standard Gaussian computations. The case for $z_2$ is analogous, which establishes that Assumption 1.1 holds with $\gamma_{\text{ntk}} \geq \sqrt{1 - \gamma_0^2}/16$, which implies the result by applying Theorem 2.1, using the simplification $\sqrt{1 - \gamma_0^2} \geq 1/2$ since $\gamma_0 \leq 1/4$, and lastly obtaining convergence to KKT directions via (Lyu & Li, 2019; Ji & Telgarsky, 2020a). □

## D    PROOFS FOR SECTION 3

This section develops the proofs of Theorem 3.2 and Theorem 3.3. Before proceeding, here is a quick sampling bound which implies there exist ReLUs pointing in good directions at initialization, which is the source of the exponentially large widths in the two statements.

**Lemma D.1.** *Let $\epsilon > 0$ and $((\alpha_k, \beta_k))_{k=1}^r$ with $(\alpha_k, \beta_k) \in \mathbb{R} \times \mathbb{R}^d$ be given with $\|\beta_k\| = 1$, and suppose $((a_j, \widetilde{v}_j))_{j=1}^m$ are sampled iid so that with $\Pr[\text{sgn}(a_j) = +1] = \Pr[\text{sgn}(a_j) = -1] = 1/2$ and $\widetilde{v}_j$ is distributed uniformly on the surface of the unit sphere in $\mathbb{R}^d$ (e.g., sample $v_j \sim \mathcal{N}_d$ and choose $\widetilde{v}_j := v_j/\|v_j\|$). If*

$$m \geq 4\left(\frac{2}{\epsilon}\right)^{d-1}\ln\frac{r}{\delta},$$

*then with probability at least $1 - \delta$, for every $(\alpha_k, \beta_k)$ there exists $(a_j, \widetilde{v}_j)$ with $\text{sgn}(\alpha_k) = \text{sgn}(a_j)$ and $\|\beta_k - \widetilde{v}_j\| \leq \epsilon$ (equivalently, $\widetilde{v}_j^\intercal \beta_k \geq 1 - \epsilon^2/2$).*

*Proof.* By standard sampling estimates (Ball, 1997, Lemma 2.3), for any fixed $k$ and $j$, then

$$\Pr[\|\widetilde{v}_j - \beta_k\| \leq \epsilon] \geq \frac{1}{2}\left(\frac{\epsilon}{2}\right)^{d-1},$$

and since all $((a_j, \widetilde{v}_j))_{j=1}^m$ are iid,

$$\Pr[\exists j \centerdot \text{sgn}(\alpha_k) = \text{sgn}(a_j) \wedge \|\widetilde{v}_j - \beta_k\| \leq \epsilon] = 1 - \Pr[\forall j \centerdot \text{sgn}(\alpha_k) \neq \text{sgn}(a_j) \vee \|\widetilde{v}_j - \beta_k\| > \epsilon]$$

$$= 1 - \Pr[\text{sgn}(\alpha_k) \neq \text{sgn}(a_1) \vee \|\widetilde{v}_1 - \beta_k\| > \epsilon]^m$$

$$= 1 - \left(1/2 + (1/2) \cdot (1 - \Pr[\|\widetilde{v}_1 - \beta_k\| \leq \epsilon])\right)^m$$

$$\geq 1 - \left(1 - (\epsilon/2)^{d-1}/4\right)^m$$

$$\geq 1 - \exp\left(-\frac{m}{4}(\epsilon/2)^{d-1}\right)$$

$$\geq 1 - \frac{\delta}{r},$$

and union bounding over all $(\beta_k)_{k=1}^r$ gives the first claim; for the alternative form, it suffices to note $\|\widetilde{v}_j - \beta_k\|^2 = 2 - 2\widetilde{v}_j^\intercal \beta_k$ and to rearrange. □

First comes the proof of Theorem 3.2, whose entirety is the construction of a potential $\Phi$ and a verification that it satisfies the conditions in Lemma B.7.

*Proof of Theorem 3.2.* The method of proof is to define a potential $\Phi$ as

$$\Phi(W) := \frac{1}{4} \sum_k |\alpha_k| \ln \sum_j \phi_{k,j} \|a_j v_j\|,$$

where (heavily dropping time indices and even the argument $w_j$ to reduce clutter)

$$\phi_{k,j}(w_j) := \phi_{k,j} := \phi\left(\widetilde{\alpha}_k a_j \sigma(v_j^\intercal \beta_k) - (1-\epsilon)\|a_j v_j\|\right),$$
$$\phi(z) := \max\{0, \min\{1, z\}\},$$

and to then verify the conditions of Lemma B.7 with $\tau = 0$ and $\widehat{\gamma} := \frac{\gamma_{\mathrm{nc}} - \epsilon}{2r}$, where the test error bound follows by Lemma C.6. By the lower bound on $m$ and Lemma D.1, it follows that $\Phi(W_0) > -\infty$, and moreover, for any $t \geq 0$, by AM-GM,

$$\Phi(W_t) \leq \frac{1}{2} \sum_k |\alpha_k| \ln \sum_j \phi_{k,j} \|w_j\|^2 \leq \frac{1}{2} \sum_k |\alpha_k| \ln \|W_t\|^2 = \frac{1}{2} \ln \|W_t\|,$$

whereby it only remains to show $\mathrm{d}\Phi/\mathrm{d}t \geq \mathcal{Q}\widehat{\gamma}/n$. To this end, note that if we could show

$$\frac{\mathrm{d}}{\mathrm{d}t} \phi_{k,j}(w_j(t)) \geq 0 \qquad \forall j, k, t, \tag{D.1}$$

then the proof is complete, since after noting the calculation

$$\begin{aligned}
\frac{\mathrm{d}}{\mathrm{d}t} \|a_j v_j\| &= \frac{\mathrm{d}}{\mathrm{d}t} \langle a_j v_j, a_j v_j \rangle^{1/2} \\
&= \frac{2 \langle a_j v_j, \dot{a}_j v_j + a_j \dot{v}_j \rangle}{2 \langle a_j v_j, a_j v_j \rangle^{1/2}} \\
&= \frac{\left\langle a_j v_j, -\sum_i \ell_i' y_i \left[ v_j \sigma(v_j^\intercal x_i) + a_j^2 \sigma'(v_j^\intercal x_i) x_i \right] \right\rangle}{n \|a_j v_j\|} \\
&= \frac{-\sum_i \ell_i' p_i(w_j) \|w_j\|^2}{n \|a_j v_j\|} \\
&= -\frac{1}{n} \sum_i \ell_i' \widetilde{a}_j \sigma(\widetilde{v}_j^\intercal x_i) \|w_j\|^2,
\end{aligned}$$

then $(\mathrm{d}/\mathrm{d}t)\Phi(W_t)$ can be lower bounded as

$$\begin{aligned}
\frac{\mathrm{d}}{\mathrm{d}t} \Phi(w) &= \frac{1}{4} \sum_k |\alpha_k| \frac{\sum_j \left[ \phi_{k,j} \frac{\mathrm{d}}{\mathrm{d}t} \|a_j v_j\| + \|a_j v_j\| \frac{\mathrm{d}}{\mathrm{d}t} \phi_{k,j} \right]}{\sum_j \phi_{k,j} \|a_j v_j\|} \\
&\geq \frac{1}{4} \sum_k |\alpha_k| \frac{-\sum_i \ell_i' y_i \sum_j \phi_{k,j} \widetilde{a}_j \sigma(\widetilde{v}_j^\intercal x_i) \|w_j\|^2}{\sum_j \phi_{k,j} \|a_j v_j\|} \\
&= \frac{1}{4n} \mathcal{Q} \sum_k |\alpha_k| \sum_{i \in S_k} q_i \frac{y_i \sum_j \phi_{k,j} \widetilde{\alpha}_k \sigma\left((\widetilde{v}_j - \beta_k + \beta_k)^\intercal x_i\right) \|w_j\|^2}{\sum_j \phi_{k,j} \|a_j v_j\|} \\
&\geq \frac{1}{4n} \mathcal{Q} \sum_k \alpha_k \sum_{i \in S_k} q_i \frac{\sum_j \phi_{k,j} (\gamma_{\mathrm{nc}} - \epsilon) 2\|a_j v_j\|}{\sum_j \phi_{k,j} \|a_j v_j\|} \\
&\geq \frac{1}{n} \mathcal{Q} \left( \frac{\gamma_{\mathrm{nc}} - \epsilon}{2r} \right);
\end{aligned}$$

the rest of the proof will establish eq. (D.1). Note moreover that eq. (D.1) has an explicit interpretation as nodes getting trapped in good directions.

Fix any $j, k, t$, and first note

$$\frac{\mathrm{d}}{\mathrm{d}t}\widetilde{\alpha}_k a_j \sigma(v_j^\mathsf{T}\beta_k) = \widetilde{\alpha}_k \left[\dot{a}_j \sigma(v_j^\mathsf{T}\beta_k) + a_j \frac{\mathrm{d}}{\mathrm{d}t}\sigma(v_j^\mathsf{T}\beta_k)\right]$$

$$= -\frac{1}{n}\widetilde{\alpha}_k \sum_i \ell_i' y_i \left[\sigma(v_j^\mathsf{T}x_i)\sigma(v_j^\mathsf{T}\beta_k) + a_j^2 \sigma'(v_j^\mathsf{T}\beta_k)x_i^\mathsf{T}\beta_k\right],$$

$$= -\frac{1}{n}\widetilde{\alpha}_k \sum_i \ell_i' y_i \left[\|v_j\|^2 \sigma(\widetilde{v}_j^\mathsf{T}x_i)\sigma(\widetilde{v}_j^\mathsf{T}\beta_k) + a_j^2 \sigma'(v_j^\mathsf{T}\beta_k)x_i^\mathsf{T}\beta_k\right],$$

whereby $(\mathrm{d}/\mathrm{d}t)\phi_{k,j} = 0$ when the argument to $\phi$ is not in $[0,1]$, and otherwise

$$\frac{\mathrm{d}}{\mathrm{d}t}\phi_{k,j} = \frac{\mathrm{d}}{\mathrm{d}t}\left[\widetilde{\alpha}_k a_j \sigma(v_j^\mathsf{T}\beta_k) - (1-\epsilon)\|a_j v_j\|\right]$$

$$= -\frac{1}{n}\sum_i \ell_i' y_i \left[\|v_j\|^2 \widetilde{a}_j \sigma(\widetilde{v}_j^\mathsf{T}x_i)\left(\widetilde{\alpha}_k \widetilde{a}_j \sigma(\widetilde{v}_j^\mathsf{T}\beta_k) - (1-\epsilon)\right)\right.$$

$$\left. + a_j^2 \left(\widetilde{\alpha}_k \sigma'(v_j^\mathsf{T}\beta_k)\beta_k^\mathsf{T}x_i - (1-\epsilon)\widetilde{a}_j \sigma(\widetilde{v}_j^\mathsf{T}x_i)\right)\right].$$

Analyzing the two bracketed terms separately, the first (the coefficient to $\|v\|^2$) is nonnegative since the term in parentheses is a rescaling of the argument to $\phi$ within $\phi_{k,j}$, which was assumed in $[0,1]$, meaning $\widetilde{\alpha}_k \widetilde{a}_j \sigma(\widetilde{v}_j^\mathsf{T}\beta_k) - (1-\epsilon) \geq 0$.

The second bracketed term (the coefficient of $a_j^2$) is more complicated. To start, fix any example $(x_i, y_i)$, define $z_i := x_i y_i$ for convenience, and define and orthogonal decomposition $z_i := c\beta_k + c_\perp z_\perp$ with $\|z_\perp\| = 1$ and necessarily $c_\perp \leq \sqrt{1-c^2}$ since $\|z_i\| \leq 1$, but more importantly $c_\perp/c \leq \sqrt{\epsilon/2}$ by Assumption 3.1. Similarly, define $\widetilde{u}_j := \widetilde{a}_j \widetilde{v}_j$ for convenience, and additionally an orthogonal decomposition $\widetilde{u}_j = q\beta_k + \sqrt{1-q^2}u_\perp$, which made use of $\|\widetilde{u}_j\| = \|\widetilde{v}_j\| = 1$, and note

$$1 - q^2 \leq 1 - (1-\epsilon)^2 \leq 2\epsilon - \epsilon^2 \leq 2\epsilon.$$

With this notation in hand, the second term becomes

$$\widetilde{\alpha}_k \beta_k^\mathsf{T}x_i y_i - (1-\epsilon)\widetilde{a}_j \widetilde{v}_j^\mathsf{T}x_i y_i = c - (1-\epsilon)\left\langle q\beta_k + \sqrt{1-q^2}u_\perp, c\beta_k + c_\perp z_\perp \right\rangle$$

$$= c\left[1 - (1-\epsilon)\left(q + \frac{c_\perp}{c}\sqrt{1-q^2}\right)\right]$$

$$\geq c\left[1 - (1-\epsilon)\left(1 + \sqrt{\epsilon/2}\cdot\sqrt{2\epsilon}\right)\right]$$

$$= c\left[1 - (1 - 2\epsilon + \epsilon^2)\right]$$

$$\geq 0,$$

as desired: $\mathrm{d}\phi_{k,j}/\mathrm{d}t \geq 0$ for every pair $(k, j)$, meaning eq. (D.1) has been established, and the proof is complete. $\qquad\square$

To close, the proof of Theorem 3.3.

*Proof of Theorem 3.3.* As in the proof of Theorem 3.2, the method of proof will be to construct a potential function $\Phi$ and then verify the conditions on Lemma B.7 with the choices $\tau = 0$ and $\widehat{\gamma} := \gamma_{\mathrm{gl}}/2 = \gamma_{\mathrm{gl}} - \epsilon$ where $\epsilon := \gamma_{\mathrm{gl}}/2$ throughout the proof, and to then apply Lemma C.6 to obtain the test error bound. Throughout the proof, use $W = ((a_j, b_k))_{j=1}^m$ to denote the full collection of parameters, even in this scalar parameter setting, and define $\widetilde{b}_j := \mathrm{sgn}(b_j)$ in mimicry of $\widetilde{a}_j$ and $\widetilde{v}_j$. To develop $\Phi$, a few other properties must first be checked.

The first property is that $a_k^2 = b_k^2$ for all times $t$; this follows directly, from the initial condition $a_k(0)^2 = b_k(0)^2 = 1/\sqrt{m}$, since at any later time $t$ it holds that

$$
\begin{aligned}
a_k(t)^2 - b_k(t)^2 &= a_k(t)^2 - a_k(0)^2 - b_k(t)^2 + b_k(0)^2 \\
&= \int_0^t \left( a_k \dot{a}_k - b_k \dot{b}_k \right) \mathrm{d}s \\
&= \frac{1}{n} \int_0^t \sum_i |\ell_i'| \left( a_k \sigma(b_k v_k^\mathsf{T} x_i) - a_k \sigma'(b_k v_k^\mathsf{T} x_i) v_k^\mathsf{T} x_i \right) \mathrm{d}s \\
&= 0.
\end{aligned}
$$

This also implies that $a_k^2 + b_k^2 = 2a_k^2 = 2|a_k| \cdot |b_k|$ throughout.

Next, for each $\beta_k$, choose $j$ so that $\|\widetilde{b}_j(0)\widetilde{v}_j - \beta_k\| \le \epsilon = \gamma_{\mathrm{gl}}/2$ and $\widetilde{a}_j(0) = \mathrm{sgn}(\alpha_k)$; this holds with probability at least $1 - \delta$ via Lemma D.1 since $\widetilde{v}_j(0)\widetilde{v}_j$ is equivalent in distribution to sampling $\widetilde{v}_j$ alone. For the rest of the proof, reorder the weights $((a_j, b_j, v_j))_{j=1}^m$ so that each $(\alpha_k, \beta_k)$ is associated with $(a_k, b_k, v_k)$. Moreover, it will be shown later in the proof that $\|\widetilde{b}_j(t)\widetilde{v}_j - \beta_k\| \le \epsilon$ and $\widetilde{a}_j(t) = \mathrm{sgn}(\alpha_k)$ in fact hold for all $t$.

Now define the potential

$$
\Phi(W) := \frac{1}{4} \sum_{k=1}^r |\alpha_k| \ln \left( a_k^2 + b_k^2 \right).
$$

Note directly that $\Phi(W_0) > -\infty$ by the above application of Lemma D.1 and choice of $((a_k, b_k))_{k=1}^r$, and moreover that

$$
\Phi(W) = \frac{1}{4} \sum_{k=1}^r |\alpha_k| \ln \left( a_k^2 + b_k^2 \right) \le \frac{1}{4} \sum_{k=1}^r |\alpha_k| \ln \|W\|^2 = \frac{1}{2} \ln \|W\|,
$$

whereby it only remains to verify $(\mathrm{d}/\mathrm{d}t)\Phi(W_t) \ge \mathcal{Q}\gamma_{\mathrm{gl}}/(2n)$.

To this end, let $T$ denote the earliest time such that $a_k(T) = 0$ for some $k \in \{1, \ldots, r\}$, which also means $b_k(T) = 0$ for that $k$ and moreover $T$ is the earliest time $b_{k'}(T) = 0$ for any $k' \in \{1, \ldots, r\}$ since $a_k^2 = b_k^2$ unconditionally for all $t$. Then, for any $t \in [0, T)$,

$$
\begin{aligned}
\frac{\mathrm{d}}{\mathrm{d}t}\Phi(W) &= \frac{1}{n} \sum_k \sum_i |\ell_i'| |y_i| |\alpha_k| \frac{a_k \sigma(b_k v_k^\mathsf{T} x_i)}{a_k^2 + b_k^2} \\
&= \frac{1}{n} \mathcal{Q} \sum_k \sum_i q_i y_i \alpha_k \frac{|a_k| \|b_k v_k\| \sigma(\widetilde{b}_k \widetilde{v}_k^\mathsf{T} x_i)}{2|a_k| \cdot |b_k|} \\
&= \frac{1}{n} \mathcal{Q} \sum_k \sum_i q_i y_i \alpha_k \sigma \left( (\widetilde{b}_k \widetilde{v}_k - \beta_k + \beta_k)^\mathsf{T} x_i \right) \\
&\ge \frac{1}{n} \mathcal{Q} \sum_k \sum_i q_i y_i \alpha_k \sigma \left( \beta_k^\mathsf{T} x_i \right) - \frac{1}{n} \mathcal{Q} \sum_k \sum_i q_i |\alpha_k| \left\| \widetilde{b}_k \widetilde{v}_k - \beta_k \right\| \\
&\ge \frac{1}{n} \mathcal{Q}\gamma_{\mathrm{gl}} \sum_i q_i - \frac{\epsilon}{n} \mathcal{Q} \sum_k \sum_i q_i |\alpha_k| \\
&= \frac{1}{n} \mathcal{Q} \sum_i q_i \left( \gamma_{\mathrm{gl}} - \epsilon \right) = \frac{\mathcal{Q}\gamma_{\mathrm{gl}}}{2n} > 0,
\end{aligned}
$$

which establishes the desired lower bound on $(\mathrm{d}/\mathrm{d}t)\Phi$ for $t \in [0, T)$, but moreover establishes (after integrating along $[0, T)$) that $\Phi(W_T) \ge \Phi(W_0) > -\infty$, which means there can not exist $k$ with $a_k(T) = 0$, since that would mean $b_k(T) = 0$ as well (as above) and thereby $\Phi(W_T) = -\infty$. Consequently, $T = \infty$ and $(\mathrm{d}/\mathrm{d}t)\Phi \ge \mathcal{Q}\gamma_{\mathrm{gl}}/(2n)$ holds for all $t$, and the proof is complete. $\qquad\square$

