# OpenReview forum: "Feature selection and low test error in shallow low-rotation ReLU networks"
_ICLR.cc/2023/Conference — ICLR 2023 poster_

### Official Review · Reviewer_Wf2D · 2022-10-24

**Confidence:** 3
**Correctness:** 2
**Technical Novelty And Significance:** 3
**Empirical Novelty And Significance:** Not applicable
**Recommendation:** 6

**Clarity, Quality, Novelty And Reproducibility:**

Clarity:
- The paper is overall clearly written and easy-to-follow.

Quality:
- The paper quality is overall good. The results are clearly stated, proof ideas are discussed, and full proofs are provided in the appendix.
- As mentioned in the weakness part, I have one concern about the proof.

Novelty:
- The theoretical results seem to be new and novel. Several interesting proof ideas/techniques are discussed.

Reproducibility:
- This is a theoretical work so there are no experiments to reproduce. Some proof ideas are discussed in the main text and full proofs are given in the appendix. I didn’t carefully check them.


**Strength And Weaknesses:**

Strength:
- The paper is overall well-written and easy-to-follow.
- Understanding the optimization and generalization of neural networks with finite samples is a very important problem in deep learning theory, and current paper studies it in the two-layer ReLU networks with low-rotation of the first layer and gives some interesting results.
- Some of the interesting proof ideas/techniques are presented, which uses the idea of margin. These might be of independent interest.

Weaknesses:
- While the result in Corollary 2.2 is interesting as it shows that gradient flow could escape bad KKT point in certain cases, the example here seems to be somewhat artificial in the sense that the dimension is only 2 and the labels are all +1. Given that this is a classification problem, it is a bit strange to me that the data have the same label.
- Concerns about the correctness of Theorem 2.1 and 2.3

    First, there is a mismatch of the initialization scale of the second layer $a$ between $a\sim N_m /\sqrt{m}$ in section 1.1 (page 2) and $a\sim N_m$ in Assumption 1.2. Under the initialization scale in Assumption 1.2, Proposition 1.3 indeed holds. Then, Proposition 1.3 was directly used in Lemma A.3. However, there seems to be an issue here: I believe Lemma A.3 in fact uses the initialization scale of $N_m/\sqrt{m}$ instead of $N_m$ because later Lemma A.3 was directly used in the proof of Theorem 2.1 and 2.3 which use the fact that $a\sim N_m/\sqrt{m}$. If the above is true, then after fixing such issue, the lower bounds in the RHS of Lemma A.3 should be scaled down by a factor of $\sqrt{m}$, which I believe would greatly change the results of Theorem 2.1 and 2.3. I would greatly appreciate if authors could comment on this to clarify my concern, and please correct me if I misunderstood any part of the proof.
- The results for the networks with large width require exponential in $d$ (input dimension) number of neurons, which makes the result not very satisfying.

Minor:
- In the line 5 of the second paragraph of page 6, $\gamma$ should be $\gamma_{ntk}$


**Summary Of The Paper:**

In this paper, the authors studied the optimization and generalization of two-layer ReLU network when the first layer neuron rotate little. Three regimes are considered, including the near initialization regime, Neural Collapse regime and non-rotate regime. The results show that low test error could be achieved and have some improvements over prior works in terms of either sample complexity or the width of networks. Several interesting proof ideas/techniques are discussed. Numerical experiments are also provided to support the theoretical results.

**Summary Of The Review:**

In summary, this work studied the problem of using GF and SGD on two-layer ReLU network in the classification setting and focus on the regime where the first layer neurons do not rotate much. The results show that a low test error could be achieved under different assumptions. Some of the proof ideas/techniques are also interesting. However, as mentioned in the weakness part, I currently have the concern of some part of the proof. Therefore, I’m currently leaning towards rejection. I’m willing to increase my score if authors could clarify my concerns.

---

> ### Author Response · Authors · 2022-11-19
> **Thank you**
>
> We thank the reviewer for their comments, time, and support.  As follows are our main responses, though we have also made revisions based on all reviewer remarks, even if we neglected to explicitly address them here.
>
> 1. **"While the result in Corollary 2.2 is interesting, [...] it is a bit strange to me that the data have the same label."**  Thank you for pointing this out; in the new appendix subsection D.5 we have provided a new construction which is obtained by essentially welding together two copies of the old construction, but now in $3$ dimensions and using two different labels.  In fact, this new construction (depicted in the new Figure 6) can be seen as a perturbed $2$-sparse parity instance, and therefore constitutes an interesting point of comparison to the other parts of the work.  If the reviewer likes this new construction and/or has any other comments, we will gladly move it to the body, replacing the original construction.
>
>
> 1. **"The results for the networks with large width [...] not very satisfying."** We agree, and in fact, due to a response to reviewer kiWi, have conducted experiments in the new appendix subsection D.4 which make i clear that a narrow network should suffice.  Our plan is to include a refinement of this table and a further discussion of narrow width in the concluding open problems in further revisions.  We also note, as in the text that prior work on global margin maximization had infinite width, and prior work on neural collapse did not provide convergence analyses from initialization.
>
> 1. **"Concerns about the correctness of Theorem 2.1 and 2.3".** We thank the reviewer for identifying this mistake.  Unfortunately, we thought it was merely a rescaling typo and devoted the revision period to many other reviewer requests (see appendix D), and only just before the revision deadline did we appreciate the severity of the issue; this is clearly a compounded mistake on our part, and we appreciate any patience and consideration the reviewer is willing to spare here.  As follows is the current status:
>
>     - We agree the issue is quite serious, and a direct fix seems to shrink the margin significantly, which in turn could preserve some of the results, but break the large margin guarantees, the good KKT point guarantees, and the property of exiting the NTK, which are the unique properties of the proofs.
>
>     - As such, we have opted for a rather drastic fix where we have changed the initialization to be balanced, meaning the outer weights have coordinates with standard deviation $1/m^{1/4}$, and similarly the inner weights have standard deviation $1/(m^{1/4} \sqrt{d})$.  We have used this fix to redo the core lemmas that exhibited the bug (the use of two different initialization scales); see in particular the new appendix subsection D.6.
>
>     - We believe that with this fix, we can still prove the above three key new properties of our analysis, namely the large margin guarantees, the good KKT point guarantee, and the NTK exit.  However, we ran out of time typing this for the revision deadline.  Instead, as we can still place comments here, we will respond to this comment by the end of November 19, AoE, with a brief explanation of how the other near-initialization guarantees change in the presence of this corrected limit and adjusted initialization scale.
>
>     We greatly appreciate that the reviewer found this mistake and look forward to their comments on the full fix.
>
>
> We thank the reviewer once again.

---

> > ### Author Response · Authors · 2022-11-20
> > **Thank you for your patience**
> >
> > We thank the reviewer for their time; here is how the near-initialization guarantees and proofs change when using the new Lemma D.5 from the revision (the bugfix of the old Lemma A.3), or a slight modification of Lemma D.5.  We also recall that we are now using a balanced initialization for these revised near-initialization proofs.
> >
> > 1. **Using the new Lemma D.5 verbatim, leading to a worse width.**
> >
> >     - Theorem 2.1, the near-initialization gradient flow theorem, now needs $m$ at least on the order of $1 / \gamma_{\textrm{ntk}}^{10}$, and achieves a margin $\gamma_{\textrm{ntk}}^7$, which means a term $1/\gamma_{\textrm{ntk}}^{14}$ in the margin-based generalization bound.  In particular, for 2-sparse parity, the width needed is $d^{10}$, which is worse than prior work as listed in Table 1.
> >
> >     - Corollary 2.2, establishing convergence to non-trivial KKT points, still goes through qualitatively (still establishes reaching non-trivial KKT points), though quantitatively the numbers worsen, since Theorem 2.1 has worsened.  The construction, however, stays the same.
> >
> >     - Theorem 2.3, the SGD guarantee, requires a width at least $1 / \gamma_{\textrm{ntk}}^{10}$ as with the revised gradient flow proof, however the sample complexity manages to remain $1/(\epsilon \gamma_{\textrm{ntk}}^2)$ as before the bugfix, and the distance to initialization after one gradient step now becomes $\gamma_{\textrm{ntk}}^8 \delta^4 m^{1/4} / (10 \cdot 2^{35})$, which while degraded, still constitutes an exit to the NTK (since the guarantee holds for any sufficiently large width $m$, and in particular it can be made arbitrarily large for this distance lower bound).
> >
> >     - **Summarizing,** qualitatively the results go through, in the sense that good margins are achieved, non-trivial KKT points are reached, and SGD still exits the NTK and has good sample complexity, however quantitatively the width is worse (and in particular worse than prior work).
> >
> >     - **Regarding the proofs,** they all basically go through identically to the old proofs after plugging in the new Lemma D.5.  A few quantities need to be chosen, but this is not too hard; to achieve the results above, it sufficed (e.g., within the SGD proof; the quantities in other proofs are the same) to choose $R$ on the order of $\gamma_{\textrm{ntk}}^6 m^{1/4}$ and $\eta$ on the order of $\gamma_{\textrm{ntk}}^7$, in particular still a constant step size.
> >
> > 2. **Using a slight modification of the new Lemma D.5, nearly recovering the old guarantees, including their width.**
> >
> >     - **A slight modification to the new Lemma D.5:** the critical term leading to the large width is the appearance of $(R^{2/3} m^{11/6})^{1/4}$; by re-ordering the factorization and Cauchy-Schwarz steps leading to this term, it is possible to obtain he term $R^{1/3} m ^{5/12}$, which may not seem like much, but will be enough to beat the $d^{8}$ width on 2-sparse parity from prior work.
> >
> >     - Referencing the preceding summary, now the width needed in Theorems 2.1 and 2.3 is on the order $1 / \gamma_{\textrm{ntk}}^{16/3}$, which in particular for 2-sparse parity leads to $d^{16/3}$ which while larger than $d^2$ listed in Table 1, is still significantly better than $d^8$ from prior work, and $d^{10}$ listed above.  As a second improvement, Theorem 2.1 achieves a margin $\gamma_{\textrm{ntk}}^4$, whereas above $\gamma_{\textrm{ntk}}^7$ was achieved.  Corollary 2.2 is qualitatively the same, though there is a slight quantitative improvement.
> >
> >     - Regarding how the proofs of Theorem 2.1, Corollary 2.2, and Theorem 2.3 change, just as in the preceding situation (using Lemma D.5 directly) one need only calculate a few new constants and everything else is fairly modular; the new improvement is that we can choose $R$ on the order $\gamma_{\textrm{ntk}}^3 m^{1/4}$ and $\eta$ on the order $\gamma_{\textrm{ntk}}^4$.
> >
> > As an overall summary, if one accepts the balanced initialization and the bugfix in the second item above, then all qualitative improvements arising in the original near-initialization proofs are preserved: the width needed for 2-sparse parity from prior work is beaten, SGD exits the NTK, GF achieves large margins, and GF can reach non-trivial KKT points.
> >
> > Is the reviewer satisfied with this bugfix?  Are there any further details which deserve clarification?
> >
> > Of course, the balanced initialization is not ideal.  On the other hand,  the four items we just mentioned were open with any initialization (i.e., a general NTK-exit for SGD, large margin GF, and GF reaching good KKT points).
> >
> > We apologize once again for not fully appreciating this issue in time for the revision deadline.  The reviewer kindly stated that our paper is clearly written; we would follow the same clarity when incorporating these fixes.
> >
> > We thank the reviewer once again for their time and for finding our mistake.

---

> ### Author Response · Authors · 2022-11-27
> **Thank you for your time.**
>
> Dear Reviewer Wf2D,
>
> Have you had a chance to look at our bugfix?  Would you like any further detail or commentary?
>
> We appreciate your time very much and hope you are satisfied with our work.  Thank you for performing a close technical read and identifying a gap in the original submission.

---

> > ### Comment · Reviewer_Wf2D · 2022-11-28
> > **Thanks for the response**
> >
> > Hi,
> >
> > I go through the new Lemma D.5 and feel that it is a reasonable fix. The new results outlined by the authors using new Lemma D.5 also make sense to me (though I do not do the calculations to check the orders of the quantities). Therefore, I'm willing to increase my score. Authors please make sure to update the proof as well as the results in the revision.
> >
> > Thanks

---

> > > ### Author Response · Authors · 2022-11-29
> > > **Thank you.**
> > >
> > > We greatly appreciate the time you spent finding our mistake, and confirming our fix.  We will work hard to incorporate this modification; we are proud of our work and grateful to the time and consideration you have given it.  As elsewhere, there are many other improvements we will incorporate, for instance the more realistic KKT construction you motivated; this construction can also be used to demonstrate our neural collapse setting and definitions.
> > >
> > > Thank you!

---

### Official Review · Reviewer_FAxZ · 2022-10-24

**Confidence:** 3
**Correctness:** 4
**Technical Novelty And Significance:** 3
**Empirical Novelty And Significance:** Not applicable
**Recommendation:** 8

**Clarity, Quality, Novelty And Reproducibility:**

The proof techniques are novel. The presentation is confusing and could be improved.

Question:

Does the true distribution also satisfy the neural collapse condition (Assumption 3.1) in Theorem 3.2? Assumption 3.1 appears to be described only for training dataset $(x_i,y_i)_{i=1}^n$.

**Strength And Weaknesses:**

**Strengths**:

- The improvement of the required network width (Theorem 2.1, 2.3, and 3.3) over existing results is good.
- This work develops several proof techniques: (i) tools to analyze low-width networks
near initialization, (ii) a new generalization bound technique, and (iii) a new potential function technique for global margin maximization far from initialization. These techniques will be helpful for future work.

**Weaknesses**:

The presentation is somewhat involved.

- Although the standard margins (normalized margin and normalized smoothed margin) are introduced, they are not used in the main text. Thus, the reasons why these were introduced are unclear, which may confuse the reader.
- The explanation of the potential function on page 8 is abrupt and difficult to understand. To help readers understand the benefit of the potential function, it would be better to first explain the motivation for using it.

**Summary Of The Paper:**

This paper studies the test error of the gradient flow (GF) and stochastic gradient descent (SGD) for two-layer ReLU networks. The authors provide several improved results over existing studies under several margin conditions for classification problems. Specifically, the required network width is significantly improved from $d^8$ to $d^2$ (Theorem 2.1 and 2.3) under NTK-margin assumption and from $\infty$ to $d^{d/2}$ (Theorem 3.3) under the global max margin assumption. Moreover, the lower sample complexity (Theorem 3.2) is shown under the neural collapse condition.

**Summary Of The Review:**

This paper makes a certain contribution in the context by improving existing results, but there is room for improvement in presentation.

---

> ### Author Response · Authors · 2022-11-19
> **Thank you**
>
> We thank the reviewer for their comments, time, and support.  As follows are our main responses, though we have also made revisions based on all reviewer remarks, even if we neglected to explicitly address them here.
>
> 1. **"The standard margins [...] are not used in the main text."**  May the reviewer clarify this point?  Theorem 2.1 includes unnormalized smoothed margins and normalized hard margins, whereas Theorems 3.1 and 3.2 include normalized smoothed margins.
>
> 1. **"The explanation of the potential function on page 8 is abrupt and difficult to understand."**  We thank the reviewer for pointing this out; as space is tight, we have left the discussion as is, but if the paper is accepted, we will have more space and can expand the discussion.  We are confident we can produce such a discussion, because the potential in fact has a clear and explicit meaning: it simply tracks how weights get trapped on good directions.  Unfortunately, there are many technicalities to deal with, which leads to the complicated nature of the final potential.
>
> 1. **"Assumption 3.1 appears to be described only for training dataset $(x_i,y_i)_{i=1}^n$."**  Thank you, we have corrected this by rewriting Assumption 3.1 to be over the support of the data distribution and not a finite sample, thus matching Assumptions 1.1 and 1.2.
>
> 1. Lastly, while not a response to a comment by the reviewer, we note that reviewer Wf2D identified a mistake in the near-initialization analysis; we have included a partial bugfix (in the appendices), and will complete the bugfix by the end of November 19 AoE in a response to Wf2D. We apologize for this oversight and are grateful for any patience and consideration.
>
> If the reviewer is satisfied with our responses but unwilling to increase their score, could they please provide further feedback?
>
> We thank the reviewer once again.

---

> > ### Comment · Reviewer_FAxZ · 2022-12-01
> > **Thanks**
> >
> > Thanks for your explanation. I read through the paper again and understood the usefulness of the potential. We hope that the revised version will include this explanation in the main text. Thus, I would like to rase the score.Thanks.

---

> > > ### Author Response · Authors · 2022-12-02
> > > **Thank you.**
> > >
> > > Thank you for your time and consideration.  We believe there is a nice way to present the potentials: we say it in words as before (trapping mass in good directions), then give a simplified mathematical version which doesn't quite do the job due to numerous technicalities, then explain the technicalities and give the final potentials.  We will have space for this.  We are proud of our work and look forward to improving it in this and other ways, thanks to all the detailed feedback from you and the other reviewers.  Thank you once again!

---

### Official Review · Reviewer_kiWi · 2022-10-24

**Confidence:** 4
**Correctness:** 4
**Technical Novelty And Significance:** 4
**Empirical Novelty And Significance:** 3
**Recommendation:** 8

**Clarity, Quality, Novelty And Reproducibility:**

The paper is overall clear and well written. The related work is adequately discussed and the novelty of the proposed approaches both in terms of results and of techniques is clear. The paper is sufficiently original to meet the acceptance bar.

Minor clarity issues:

* Figure 1 is not particularly clear. Is there any way to highlight what is the direction of the trajectories (as $t$ grows)?

* In the discussion of the proof of Theorem 2.1 in page 6, the authors mention that "the same inequality holds with $W_0$ replaced by any $W$ s.t. $||W-W_0||\le \gamma\sqrt{m}$. What's the relationship between this $\gamma$ and $\gamma_{\rm ntk}$?

* Page 24, the last inequality of the proof in Section B.1 is incomplete.

* Lemma B.5, first inequality in the statement. $||a||\cdot ||V$ should be $||a||\cdot ||V||$. Also the authors should clarify the notation $| \mathcal X$.



**Strength And Weaknesses:**

Strengths:

(1) Theorem 3.3 improves in the network width over (Ji & Telgarsky, 2020b) and in the number of samples over (Barak et al., 2022).

(2) Interesting technical innovations over existing work.

(3) The assumption on low rotation, although strong, does allow for stronger results (achieving the global margin) and seems to be satisfied experimentally.

(4) [Minor, but still nice to see...] Well written paper with the various assumptions properly discussed, the related work discussed in details, and interesting open problems.

Weaknesses:

(1) Model still somewhat artificial.

(2) The optimality of the results remains unclear. Theorem 2.3 leads to a bound on the network width of order $d^2$. This follows from having the width scaling as $1/\gamma_{\rm ntk}^2$ and then showing that $\gamma_{\rm ntk}$ scales at least linearly in $1/d$. Is any of these two bounds optimal (in some sense)? For the regression setting, the minimum width has to scale linearly with the number of parameters (this follows from counting degrees of freedom, or from VC-dimension bounds). However, for classification, it is not entirely clear what the minimum width should be. I would encourage the authors to discuss in more detail this point, which is only briefly mentioned as the open problem of understanding 'the Pareto frontier of width, samples, and computation'.

(3) In the statement of Theorem 2.1, the authors claim that there exist $t$ with $||W_t-W_0||=\gamma_{\rm ntk}\sqrt{m}/32$. Did the authors intend to put an upper bound here?

(4) Is the probability bound of Theorem 2.1 uniform in $s$?

**Summary Of The Paper:**

This paper considers shallow neural networks trained with gradient methods (stochastic gradient descent or gradient flow) under logistic or exponential loss. The authors prove that a low test error can be achieved in a regime that still exhibits low rotations in the weights, but is capable of exiting the NTK regime (in which the weights do not move much from their initialization).

More specifically, the authors consider networks with moderate width (order of $d^2$) and prove that a margin scaling linearly with $\gamma_{\rm ntk}$ can be achieved by either stochastic gradient descent (Theorem 2.3) or gradient flow (Theorem 2.1). This improves upon the width requirement in an earlier work by Ji & Telgarsky, (2020b) and upon the number of samples needed by an earlier work by Barak et al., (2022) -- although reaching the Pareto frontier of width, samples and computation remains a (possibly rather difficult) open problem.

Next, the authors consider a setting close to the 'neural collapse' (NC) popularized by the recent paper by Papyan et al., (2020). Here, the networks widths are impractical (exponential in $d$), but the authors can show a margin scaling linearly with the *global* margin $\gamma_{\rm gl}$ -- as opposed to the previous results scaling with $\gamma_{\rm ntk}$. These results also require either an assumption on a NC property of the dataset (Theorem 3.2) or on the inability of the inner weights to rotate (Theorem 3.3).

**Summary Of The Review:**

Overall, the paper contains a number of new results fuelled by new analysis ideas. The weaknesses mentioned above are overcome by the strengths, and I am positive about the submission.

---

> ### Author Response · Authors · 2022-11-19
> **Thank you**
>
> We thank the reviewer for their comments, time, and support.  As follows are our main responses, though we have also made revisions based on all reviewer remarks, even if we neglected to explicitly address them here.
>
> 1. **"The optimality of the results remains unclear"**, and associated remarks.  As a first comment, in the NTK regime but training only the inner layer, there is a specialized setting resembling $2$-sparse parity with a width lower bound of roughly $1/\sqrt{\gamma_{\textrm{ntk}}}$ (See proposition 5.4 in "polylogarithmic width [...]" by Ji and Telgarsky), but of course our paper exits the NTK, even in section 2.  Motivated by the reviewer's comments, we ran experiments trying to plot the suggested "pareto frontier" trading off between, width, computation (number of gradient steps), and samples.  To our surprise, as in the new table in the new appendix subsetion D.4, there doesn't really seem to be an interesting trade-off curve, but instead it seems a best small width and small number of samples can be simultaneously achieved.  We hope the reviewer finds this preliminary experiment interesting; we plan to include a refined version in a revised paper body.
>
> 2. **"In the statement of Theorem 2.1, [...] Did the authors intend to put an upper bound here"** and relatedly **"Is the probability bound of Theorem 2.1 uniform in $s$?"**  Regarding the first question, this is not a typo, we need exactly this $t$ and to be far from initialization so that the margin is not dominated by initial random weights.  In particular, we can prove that we have a good margin at time $t$; after that point, we can use the prior work of Lyu&Li to ensure that margins are nondecreasing, which ensures good margins and good generalization for all $s\geq t$.  In particular, we do not need any complicated argument to handle $s\geq t$ uniformly, it comes for free from a single margin bound and the margins being maintained; we have adjusted the wording of Theorem 2.1 so that this uniformity in $s$ is clear.
>
> 3. **"Figure 1 is not particularly clear."**  In the GELU experiments in the new subsection D.2, we have added markers to the ends of trajectories, and also markers partway through long trajectories (and sometimes both).  We have left the body figures as they are so they are available for comparison.  If the reviewer thinks this constitutes good progress, we can continue to refine the figures; another thing we will try is putting arrowheads halfway along each trajectory.
>
> 4. **"What's the relationship between this $\gamma$ and $\gamma_{\textrm{ntk}}$?"**  It should have been $\gamma_{\textrm{ntk}}$; we have corrected this, thank you for pointing it out.
>
> 5. **"Also the authors should clarify the notation $|\mathcal{X}$."** Thank you, we have expanded the material before our Rademacher bound with a discussion of Rademacher complexity and an explanation of this notation.
>
> 6. Lastly, while not a response to a comment by the reviewer, we note that reviewer Wf2D identified a mistake in the near-initialization analysis; we have included a partial bugfix (in the appendices), and will complete the bugfix by the end of November 19 AoE in a response to Wf2D. We apologize for this oversight and are grateful for any patience and consideration.
>
> We thank the reviewer once again.

---

> > ### Comment · Reviewer_kiWi · 2022-12-01
> > **Thoughtful response and revision**
> >
> > I am satisfied with the response and I keep my positive rating. As for the new Figure 1, I would suggest to place in the body the version currently contained in the appendix.
> >
> > Finally, kudos to reviewer Wf2D for identifying a bug and to the authors for fixing it (although at the expense of slightly weaker results and a less natural initialization). My suggestion to the authors would be to keep the balanced initialization for the final version as it allows to almost recover the original claims.

---

### Official Review · Reviewer_ECNr · 2022-10-30

**Confidence:** 3
**Clarity, Quality, Novelty And Reproducibility:** See strength and weaknesses.
**Correctness:** 4
**Technical Novelty And Significance:** 3
**Empirical Novelty And Significance:** Not applicable
**Recommendation:** 6

**Strength And Weaknesses:**

I find this submission fun to read. Despite the highly technical/theoretical content, the authors provide nice intuitive explanations of the analysis in the main text (e.g. on the construction of potential function). Also, the limitations of each approach and comparison to prior results are carefully discussed. I believe the following contributions are meaningful:

- The NTK margin analysis allows the parameters to move up to $\mathcal{O}(\sqrt{m})$ and improves the required width in terms of the NTK margin. It is also interesting that the margin-based Rademacher complexity bound has no dependence on the network width (due to manipulation of the homogeneity property).
- To my knowledge the scalar gradient flow analysis is novel. It outlines a new assumption (low rotation) that leads to global margin maximization, which differs from prior results that assume noisy dynamics or dual convergence. Moreover, this low-rotation assumption seems to hold empirically when the network width is large.

The weaknesses of this submission are also clear (as acknowledged by the authors). If we focus on the sample complexity of learning the 2-sparse parity function, then the improvement over existing results is not very significant. While the perceptron analysis is not an NTK proof, the resulting sample complexity does not show any advantage over the NTK model. As for the scalar gradient flow, the required model width still exponential even under this rotation-free parameterization.

I have the following questions/comments.

1. The authors used the 2-sparse parity problem as an illustrative example throughout the paper. What is the motivation of using this particular problem? How would the margin values change if we generalize to $k$-parity? And is the sample complexity still better than kernel methods?

2. Is there an intuitive explanation of why the large-width analysis can only handle the exponential loss?

3. Is the empirically-observed low-rotation property specific to the ReLU (or homogeneous) activation function?

4. A few more related works: the learning of parity-like functions using two-layer neural networks has been studied in [1] [2]. The two-phase algorithm (learning the features and then the second layer) has been analyzed in the regression setting in [3] [4].
[1] Refinetti et al. Classifying high-dimensional gaussian mixtures: where kernel methods fail and neural networks succeed.
[2] Frei et al. Random feature amplification: Feature learning and generalization in neural networks.
[3] Ba et al. High-dimensional asymptotics of feature learning: how one gradient step improves the representation.
[4] Bietti et al. Learning single-index models with shallow neural networks.

**Summary Of The Paper:**

This submission studies the gradient descent/flow dynamics of a two-layer neural network with ReLU activation function and losses with exponential tail. The authors presented the following results: (i) an improved perceptron analysis based on the NTK margin that allows the parameters to move away from the NTK regime, and it is shown that a polynomial-width neural network achieves low test error; when specialized to the 2-parity problem, this gives learning guarantees that matches kernel methods.
(ii) under a neural collapse assumption, an exponential-width neural network with exponential loss can achieve large margin and low test error. (iii) in a scalar gradient flow where the neurons cannot rotate, it is shown that an exponential-width neural network can find the global max-margin solution (up to constant); for the 2-parity problem, this implies superiority over kernel methods.

**Summary Of The Review:**

I am not very familiar with the max-margin literature, and I did not read the Appendix in details.
However, I believe the results presented in this submission are relevant to the ICLR community.

---

> ### Author Response · Authors · 2022-11-19
> **Thank you**
>
> We thank the reviewer for their comments, time, and support.  As follows are our main responses, though we have also made revisions based on all reviewer remarks, even if we neglected to explicitly address them here.
>
>
> 1. **"The authors used the 2-sparse parity problem [...]."**  There were many important questions here and we address them in turn.
>
>     - **"What is the motivation [...]?"**.  (a) As this problem is well-studied, it can be used to compare papers and techniques, even outside neural networks. (b) As the kernel lower bounds are known, it gives a concrete and meaningful (sample complexity and test error) goal to beat.
>
>
>     - **"How would the margin values change if we generalize to k-parity? And is the sample complexity still better than kernel methods?"**  We have calculated the $k$-bit parity margin explicitly: as in the new Proposition D.1 in the revisions, it is $1/ (2k\sqrt{d})$, which is in fact better than we expected, and much better than the $1/d^{k/2}$ which would match the kernel method sample complexity from other work.  We have also begun working out the NTK margin, and if the reviewer thinks this material is valuable, would include both in the body in further revisions.  We must admit we were surprised by how good this margin is (no exponential dependence on $k$), and therefore computer checked it as well.  We think this constitutes a valuable improvement to the paper and thank the reviewer for motivating us to investigate it.  We are happy to investigate further questions on $k$-bit parity as well.
>
>
> 2. **"Is there an intuitive explanation of why the large-width analysis can only handle the exponential loss?"**  This was an unnecessary weakness of our submission and we have revised our proofs to handle the logistic loss as well.  This ended up being easier than we expected, it was a combination of elementary calculations in lemmas in appendix subsection A.4, and also some lemmas from the prior work by Ji&Telgarsky, which can handle the logistic loss.
>
>
> 3. **"Is the empirically-observed low-rotation property specific to the ReLU (or homogeneous) activation function?"** We re-ran our experiments with the GELU and collected them in the new appendix subsetion D.2.  We chose the GELU because while it has some similar characteristics to the ReLU, in that it is "asymptotically homogeneous", but near the origin it is not only highly non-homogeneous, but moreover non-monotone, so we expected very different behavior near initialization.  Overall we found these experiments much more brittle than the ReLU experiments, but more investigation is needed.  We could have used the logistic as a smooth activation as well, but that is too similar to the ReLU in our eyes; meanwhile, the sigmoid would be drastically different (does not behave approximately $2$-homogeneous when far from the origin.  That said, we find these behaviors interesting and are happy to produce further experiments of this type (they required changing one line of code).
>
> 4. **"A few more related works[...]".** These are great papers and we appreciate the suggestion and apologize for the omission, and we agree that including these works will improve the paper.  We added a temporary appendix subsection D.3 where we discuss these missing works, but will incorporate it in the paper body in future revisions.
>
> 5. Lastly, while not a response to a comment by the reviewer, we note that reviewer Wf2D identified a mistake in the near-initialization analysis; we have included a partial bugfix (in the appendices), and will complete the bugfix by the end of November 19 AoE in a response to Wf2D. We apologize for this oversight and are grateful for any patience and consideration.
>
> We thank the reviewer once again.

---

### Decision · Program_Chairs · 2023-01-20

**Decision:**

Accept: poster

**Justification For Why Not Higher Score:**

An interesting paper with somewhat novel results but not fundamentally different from prior work.

**Justification For Why Not Lower Score:**

This is still clearly an accept, no reviewer argues for rejection

**Metareview: Summary, Strengths And Weaknesses:**

This paper studies the gradient flow/descent dynamics of a two-layer neural network with ReLU activation function and losses with an exponential tail. They derive new results for different regimes (near initialization, neural collapse, constrained layer weights that can not rotate).

The reviewers appreciate the technical novelty of the paper, although they also commented on the fact that the analysis does not provide fundamentally different results from the typical NTK analysis. Yet, there are some improvements (for instance in the terms of the requirements on the width of the network) provided by some of the theorems so this seems to be a worthwhile publication.

One reviewer identified a problem in one of the proofs which was later fixed by the authors.

Overall, the paper definitely has some interesting results and I therefore recommend acceptance.

**Note From Pc:**

if the above contains the word "oral" or "spotlight" please see: "oral" presentation means -> notable-top-5% and "spotlight" means -> notable-top-25%. As stated in our emails, we are disassociating presentation type from AC recommendations

**Summary Of Ac-Reviewer Meeting:**

N/A